# 3DLLM-Mem: Long-Term Spatial-Temporal Memory for Embodied 3D Large Language Model

Wenbo Hu[1✉]    Yining Hong[1]    Yanjun Wang[1]    Leison Gao[1]    Zibu Wei[1]
Xingcheng Yao[1]    Nanyun Peng[1]    Yonatan Bitton[2]    Idan Szpektor[2]    Kai-Wei Chang[1]

[1]University of California, Los Angeles, [2]Google Research

https://3dllm-mem.github.io

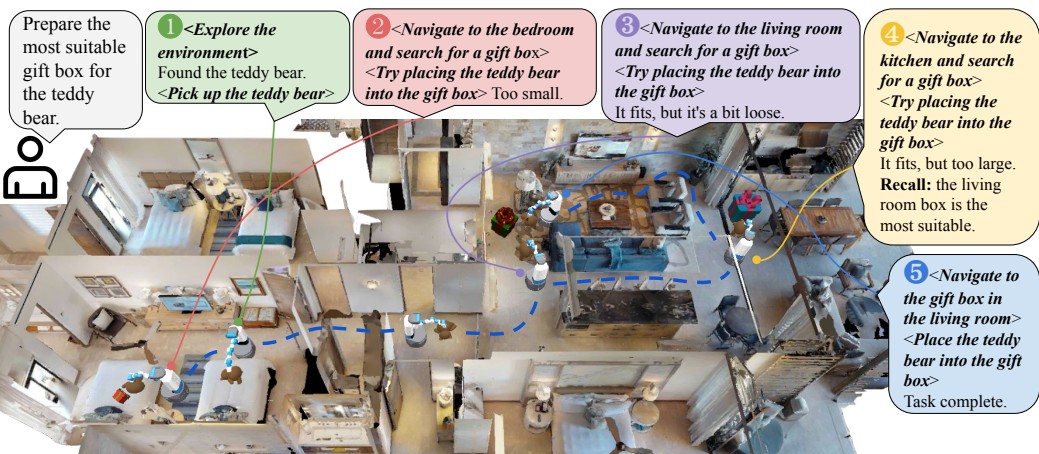

Figure 1: We propose 3DLLM-MEM, a memory-enhanced 3D embodied agent that explores and incorporates feedback from the environment, interacts with objects, and incrementally builds and maintains a task-relevant long-term memory throughout its trajectory. For illustration purposes, agents from multiple time steps are shown simultaneously.

## Abstract

Humans excel at performing complex tasks by leveraging long-term memory across temporal and spatial experiences. In contrast, current Large Language Models (LLMs) struggle to effectively plan and act in dynamic, multi-room 3D environments. We posit that part of this limitation is due to the lack of proper 3D spatial-temporal memory modeling in LLMs. To address this, we first introduce 3DMEM-BENCH, a comprehensive benchmark comprising over 26,000 trajectories and 2,892 embodied tasks, question-answering and captioning, designed to evaluate an agent's ability to reason over long-term memory in 3D environments. Second, we propose 3DLLM-MEM, a novel dynamic memory management and fusion model for embodied spatial-temporal reasoning and actions in LLMs. Our model uses *working memory* tokens, which represents current observations, as queries to selectively attend to and fuse the most useful spatial and temporal features from *episodic memory*, which stores past observations and interactions. Our approach allows the agent to focus on task-relevant information while maintaining memory efficiency in complex, long-horizon environments. Experimental results

✉ Contact at whu@cs.ucla.edu.

demonstrate that 3DLLM-MEM achieves state-of-the-art performance across various tasks, outperforming the strongest baselines by 16.5% in success rate on 3DMEM-BENCH's most challenging in-the-wild embodied tasks.

# 1 Introduction

Picture yourself traversing an unfamiliar home, as illustrated in Figure 1, on a mission to explore multiple rooms and evaluate various gift boxes to find the most suitable one for wrapping a teddy bear. As you navigate from room to room, your brain instinctively creates a 3D cognitive map of the environment, maintains a working memory of objects you've encountered, forms episodic memories that link observations across space and time, and plans efficient actions. This seamless integration of 3D spatial understanding, long-term memory encoding and retrieval, fluid switching between working and episodic memory, and purposeful action planning — cognitive processes that humans take for granted — remain formidable challenges for embodied AI systems today.

Recent extensions of Large Language Models (LLMs) to 3D environments have birthed 3D-LLMs (Hong et al., 2023b; Guo et al., 2023; Gu et al., 2024; Huang et al., 2024b; Xu et al., 2025a) that can perceive and reason about 3D spaces, while 3D Vision-Language-Action models (Zhen et al., 2024; Zhao et al., 2025; Intelligence et al., 2025) further incorporate the ability to plan and act within these environments. Despite these advances, several critical limitations persist that prevent models from performing the kinds of tasks described above. First, current models struggle to maintain long-term memory chains when performing complex tasks that unfold across multiple visual scenarios, such as several rooms in a house, and extended time frames. Real-world 3D physical scenes are remarkably vast and information-dense, where every detail can matter for long-horizon embodied tasks — for instance, in Figure 1, finding the most suitable gift box requires remembering all the gift boxes encountered along the way and their characteristics and interaction with teddy bear. Dense 3D representations are particularly valuable as they capture comprehensive spatial information, preserving intricate geometric relationships and environmental details that sparse or object-centric approaches might miss. However, how to accurately and efficiently store dense 3D memory remains a fundamental challenge - retrieving the entire history would overwhelm the model's context limits, while selective retrieval (Xie et al., 2024; Wang et al., 2024; Yang et al., 2025b) risks omitting critical information needed for accurate reasoning and decision-making. The second challenge resides in the entanglement of spatial and temporal memory — agents must track not only where objects are, but how they change over time through exploration and interaction. As environments evolve, maintaining coherent representations of previously seen spaces while incorporating new information continues to exceed the capabilities of current embodied AI models.

Our efforts at solving this challenge are two-fold. First, we introduce a novel benchmark for reasoning, planning and acting with long-term spatial-temporal memory in embodied environments. Our benchmark, 3DMEM-BENCH, encompasses multi-room 3D scenes from the Habitat environment, augmented with interactive objects to enable manipulation tasks across extended spatial-temporal horizons. Notably, we define fine-grained embodied tasks across varying levels of difficulty—from simple to hard—enabling deeper insight into model performance, which we believe is not addressed in prior benchmarks as shown in Table 1. Our task set spans a wide range of complexities, from straightforward object collection to challenging comparative reasoning tasks that require integrating observations across multiple rooms and time steps. Additionally, we include in-the-wild challenge tasks to evaluate the model's generalization capabilities beyond seen environments. The benchmark includes three evaluation categories: (1) embodied tasks requiring extended action sequences across multiple rooms, (2) spatial-temporal embodied question answering (EQA) that evaluates understanding of spatial relationships over time, and (3) long-term scene captioning that tests memorization of previously observed environments. Our dataset includes 26,000+ trajectory examples spanning 182+ unique scenes with an average of 18 rooms per scene.

Second, we introduce 3DLLM-MEM, a 3D embodied LLM with dynamic memory management capabilities designed specifically for embodied spatial-temporal reasoning, planning and acting. To our knowledge, we are among the first to explore dense 3D representations as memory for embodied 3D LLMs — addressing a significant gap in current research as noted in recent 3D memory studies (Yang et al., 2025b). Unlike standard approaches that rely solely on context windows (Hong et al., 2023b; Huang et al., 2024b; Zhu et al., 2024), 3DLLM-MEM implements a dual-memory system: a limited-capacity working memory for current observations and an expandable episodic

| Benchmark | #Test Tasks | #Train Trajectories | Long-term Memory | Fine-grained complexity | EQA | Captioning |
|---|---|---|---|---|---|---|
| ALFWorld (Shridhar et al., 2021) | 274 | 3,553 | ✗ | ✗ | NA | NA |
| Behavior-1K (Li et al., 2024a) | 1,000 | NA | ✗ | ✗ | NA | NA |
| VisualAgentBench (Liu et al., 2024) | 746 | 4,482 | ✗ | ✗ | NA | NA |
| EmbodiedBench (Yang et al., 2025a) | 1,128 | NA | ✗ | ✗ | NA | NA |
| **3DMEM-BENCH (ours)** | 1,860 | 26,276 | ✓ | ✓ | 865 | 167 |

Table 1: Comparison with related benchmarks. 3DMEM-BENCH focus on spatial-temporal memory through fine-grained embodied tasks and EQA that span multiple "pieces" of long-term memory, distinguishing it from prior benchmarks that typically target single-step or short-horizon reasoning. Fine-grained complexity indicates our embodied task spans from simple to medium to hard.

memory that stores past spatial-temporal information as dense 3D representations. The key innovation is our memory fusion module that actively integrates information from both memory systems based on task relevance and spatial-temporal relationships. This allows the model to leverage the benefits of dense 3D representations while mitigating their computational demands, maintaining coherent spatial-temporal understanding across extended task horizons. The fusion process preserves critical spatial relationships while accounting for their evolvement through agent interactions over time.

We evaluate popular 3D-LLMs and memory mechanisms on 3DMEM-BENCH. Experimental results demonstrate 3DLLM-MEM significantly outperforms all existing approaches in both in-domain and in-the-wild embodied tasks. Notably, while the performance of other methods drops sharply in the challenging in-the-wild setting, our method remains robust, achieving an average success rate of 32.1%—demonstrating strong generalization capabilities. As task complexity increases from simple to hard, all existing approaches degrade significantly, achieving only ~5% success rate in hard in-the-wild tasks. In contrast, 3DLLM-MEM maintains a strong performance of 27.8%, demonstrating its scalability and effectiveness in managing longer-term memory representations.

Our contributions can be summarized as below:

- We propose a novel task that requires agents to execute action chains while maintaining and utilizing long-term spatial-temporal memory.
- We construct 3DMEM-BENCH, a comprehensive benchmark comprising over 26,000 trajectories and 1,860 fine-grained long-term memory embodied tasks—ranging from simple to hard—along with question-answering tasks that target memory changes across time and space, and captioning tasks in complex 3D environments.
- We propose 3DLLM-MEM, an embodied 3D LLM with a novel memory fusion module for spatial-temporal reasoning, planning, and acting-which utilizes working memory tokens as queries to selectively fuse relevant features from episodic memory for efficient, task-aware decision-making.
- Experimental results on embodied tasks, question-answering, and captioning demonstrate that 3DLLM-MEM outperforms baselines by a large margin.

## 2 The Embodied 3D Long-Term Spatial-Temporal Memory Benchmark

### 2.1 Overview of 3DMEM-BENCH

**Design principles**  Long-term memory (Camina and Güell, 2017; Friedman et al., 2018; Zlotnik and Vansintjan, 2019) can be categorized into *explicit memory* and *implicit memory*. Explicit memory includes *semantic memory*, which stores general knowledge and facts about the world, and *episodic memory*, which consists of personal experiences that are time-stamped and context-specific. In contrast, implicit memory primarily involves *procedural memory*, such as learned skills and habits.

To comprehensively evaluate 3D long-term memory for real-world applications, we design 3DMEM-BENCH following three core task categories: embodied tasks, long-term memory EQA, and captioning. As illustrated in Figure 2, *embodied tasks* require an embodied agent to solve realistic indoor environment challenges by leveraging both implicit and explicit long-term memory. *Long-term memory EQA* tests the agent's ability to answer complex embodied questions using spatial-temporal memory. This task includes five subcategories: spatial reasoning questions, long-term object navigation, comparative reasoning, multi-room layout understanding, and semantic object counting. *Captioning* tasks involve summarizing the agent's episodic memory to highlight shared and distinctive features across experiences, enabling more informed decision-making under the current task context.

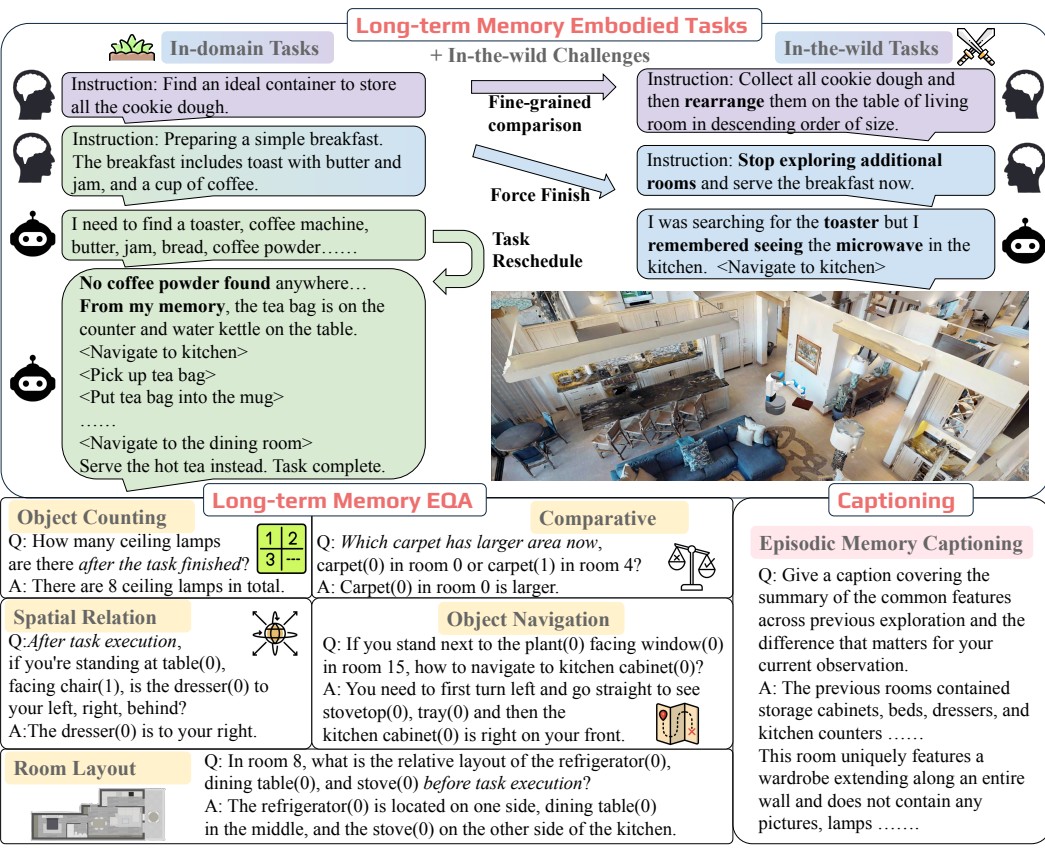

Figure 2: Overview of 3DMEM-BENCH. For long-term memory embodied tasks, we further incorporate in-the-wild challenges to test 3D agent's generalization abilities. Text inside < > indicates high-level action tokens. For complete embodied task trajectories, please refer to Appendix C.

## 2.2 Data Collection

**Base environment construction** We build our scenes on top of the Habitat-Matterport 3D (HM3D) semantics dataset (Ramakrishnan et al., 2021), which has 1000 3D spaces and 10,600 rooms within those spaces. Pre-processing for the axis-aligned bounding box and using valid semantic label annotation, we filter to 182 3D spaces and 2,602 rooms. However, existing objects in HM3D scene are not interactive in Habitat-sim (Szot et al., 2021). To expand our task diversity and enable embodied tasks, we add interactive objects from Objaverse (Deitke et al., 2023) which consists of 800K 3D objects spanning rich categories. More environment construction details are illustrated in Appendix B.

**Generating task trajectories** Following Hong et al. (2023b, 2024), we adopt box-demonstration-instruction-based prompting, which utilizes the axis-aligned bounding boxes (AABB) of both rooms and objects within the 3D scenes to prompt Gemini (Team et al., 2023) to generate diverse tasks. We further prompt Gemini to incorporate interactive objects based on task requirements and their appropriateness within indoor environments. Detailed prompt instructions and few-shot demonstration examples are provided in Appendix E. To ensure the validity of the generated trajectories, we develop a trajectory simulation pipeline that verifies each trajectory step-by-step. At every step, the simulator checks: (1) the correctness of the agent's location, (2) the existence and validity of referenced objects, and (3) the correctness of pick-up and put-down actions. Finally, we ensure that high-level actions can be executed in the simulator, following (Szot et al., 2024; Yang et al., 2025a). Details of this implementation are in Appendix F.1. On average, our filtering process yields a validation rate of approximately 24%, ensuring the correctness and feasibility of the generated trajectories.

**Embodied data collection** In our task settings, an embodied agent first performs random exploration within the environment to collect RGB-D observations and corresponding camera poses.

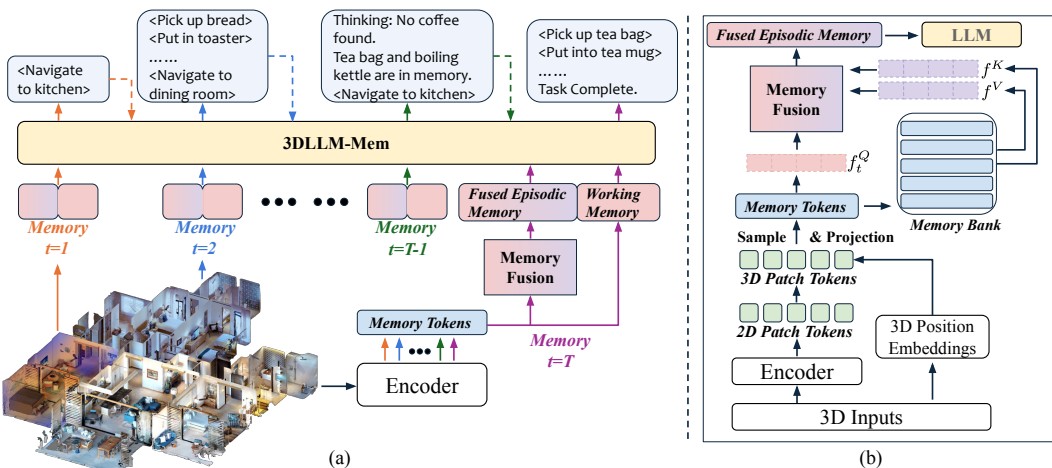

Figure 3: (a) We propose 3DLLM-MEM, a memory-enhanced 3D embodied agent that gradually form its long-term memory while executing tasks. Multiple timesteps are shown together but in different colors, with each timestep's memory including the prior one. The task is "prepare a simple breakfast" as shown in Figure 2. (b) Overview of our memory fusion mechanism.

Then the agent follows the task trajectory, incrementally exploring new environments, executing interaction actions, and receiving feedback with new RGB-D observation data. All interaction results are recorded and the reconstructed point cloud data is precomputed and stored locally to enable faster loading during both training and inference.

## 2.3 Data Curation

As mentioned previously, we collect embodied data by prompting Gemini. To enable a fine-grained analysis of long-term memory capacity, we divide the tasks into three subcategories: *simple*, *medium*, and *hard*, comprising of 3, 5 and 10 multi-room scene settings respectively. In total, we collect 51K trajectories, with 31K in the simple setting, 10K in the medium, and 10K in the hard.

To construct in-domain evaluation sets, we first remove training tasks and filter for instances that never shown in the agent's working memory. For the in-the-wild evaluation set, we apply additional filtering to assess the agent's generalization capabilities. Specifically, we select instances involving unseen objects and entirely unseen memory context, and we introduce novel in-the-wild challenges that differ from those encountered during training, as illustrated in Figure 2.

For EQA data curation, we extract complete trajectories explored by agents and then prompt Gemini to generate question-answer pairs. The questions are categorized into spatial reasoning, long-term object navigation, comparative reasoning, multi-room layout understanding, and semantic object counting. As shown in Figure 2, these questions evaluate models on spatial-temporal changes in memory during embodied task execution. For long-term memory captioning, which primarily targets semantic episodic memory, we collect data across multiple rooms before and after the execution of each trajectory, enabling comparison and summarization of memory-relevant experiences.

**Quality control** After constructing the entire benchmark, we implement two quality control procedures: automatic validation using trajectory simulation rules and a manual review of each benchmark instance. The automatic check involves re-running the trajectory simulation validation pipeline, as described in §2.2, particularly for the in-the-wild tasks. For human validation, four student experts in the field manually inspect each benchmark example. We render multi-view images of the entire scene using the simulator and verify whether the benchmark annotations accurately correspond to the simulated environment. More details are in Appendix F.2.

# 3 3D Long-Term Spatial-Temporal Memory Model (3DLLM-MEM)

## 3.1 Preliminary

Recent work on 3D Large Language Models (3D-LLMs) has showcased robust capabilities. We choose LLaVA-3D (Zhu et al., 2024) as the base model to build our long-term memory 3D-LLM.

LLaVA-3D directly builds on 2D-LLM with multi-view images as input and utilizing the 3D position embeddings to bring the 2D patches within a 3D spatial context to construct 3D patches. For each frame image, a CLIP encoder splits the image $X \in \mathbb{R}^{3 \times W \times H}$ into patches at the patch size $P$. For each 3D scene, $V$ multi-view image patch features are encoded and then projected into LLM space as $X_p \in \mathbb{R}^{V \times d \times w \times h}$, where $h = \lfloor \frac{H}{P} \rfloor$, $w = \lfloor \frac{W}{P} \rfloor$, and $d$ represents LLM's hidden dimension. The 3D positions in the 3D world are obtained with known depth image, camera intrinsic and extrinsic parameters and are further encoded into 3D position embeddings $P \in \mathbb{R}^{V \times d \times w \times h}$. These are directly added to the 2D patch visual tokens $X_p$, resulting in pixel-aligned 3D patches $X_{3D} \in \mathbb{R}^{V \times d \times w \times h}$. To reduce redundancy in 3D patches, we adopt the Farthest Point Sampling (FPS) strategy to downsample the 3D features to a fixed number of tokens, resulting in $X_{\text{3D Feat}} \in \mathbb{R}^{N \times d}$.

## 3.2 3DLLM-MEM Memory Module

A 3D embodied agent gradually explores the environment by collecting observations and interacting with surrounding environments. For humans, current observations are held in *working memory*, while longer-term observations and experiences are stored in *episodic memory*. Inspired by human cognitive structure, 3DLLM-MEM is designed with a similar paradigm as illustrated in Figure 3. The current observation at time step $t = i$, denoted as $X^{[t=i]} \in \mathbb{R}^{N \times d}$, remains within the context window and serves as the agent's *working memory*. As the agent accumulates more experiences, past observations from time steps 1 to $T$, represented as $X^{[t=1:T]} \in \mathbb{R}^{T \times N \times d}$, are stored as part of its *episodic memory*, where $T$ denotes the total number of timesteps.

**Episodic memory**    To manage episodic memory, we propose the use of a memory feature bank. For each observation at time step $j$, where $1 \leq j \leq T$, we first apply a multi-layer perceptron (MLP) layer to project the observation into a memory-specific feature space, which is then stored in the memory bank for future retrieval. To further enhance the temporal understanding of the agent's exploration, we incorporate sinusoidal positional embeddings to encode each time step $t = j$, and then directly added to the corresponding memory feature representations.

**Memory fusion**    Our motivation is that an agent should leverage its current observations to recall the most relevant information from its episodic memory in order to complete the current task. To achieve this, we propose a mechanism called *3D memory fusion*. Specifically, we encode the 3D features from the working memory into a shared memory space and use this representation as the query feature, denoted as $f_t^Q \in \mathbb{R}^{N \times M}$, where $M$ is the dimensionality of the memory feature space.

The episodic memory bank stores the corresponding key and value features from past observations: $f^K \in \mathbb{R}^{T \times N \times M}$ and $f^V \in \mathbb{R}^{T \times N \times M}$, respectively. Here, $T$ is the number of past timesteps and $N$ is the number of memory tokens per timestep. This structure allows the agent to retrieve task-relevant information through memory-query attention. The fused memory feature is then concatenated with the working memory feature to produce the final memory-enhanced representation $f^M$ for the agent:

$$f_{\text{fuse}}^Q = \text{Softmax}\left(\frac{f_t^Q (f^K)^\top}{\sqrt{C}}\right) f^V, \quad f^M = \text{Concat}\left[f_{\text{fuse}}^Q; f_t^Q\right] \tag{1}$$

**Memory update**    The working memory is dynamic and updated online. As the agent interacts with the environment, changes in the environment are immediately reflected in the working memory through updated 3D representations. When the agent moves to a new environment, the previous working memory is transferred to the episodic memory bank. If the corresponding environment already exists in the memory bank and has been modified by the agent, the memory entry is updated accordingly. Thus, the memory bank remains dynamic and reflects the latest state of the explored environments. As described in §2.2, environment changes and corresponding observations are pre-collected and stored locally to facilitate efficient data loading during both training and inference.

## 4 Experiments

In this section, we first introduce the experimental setup and existing memory management baselines in §4.1. Then, we benchmark existing approaches on 3DMEM-BENCH, and present comprehensive results on embodied tasks, EQA, and captioning tasks to demonstrate the effectiveness of our

| Model | Simple | | | | Medium | | | | Hard | | | | Average | | | |
|---|---|---|---|---|---|---|---|---|---|---|---|---|---|---|---|---|
| | In-domain | | In-the-wild | | In-domain | | In-the-wild | | In-domain | | In-the-wild | | In-domain | | In-the-wild | |
| | SR | Sub-SR | SR | Sub-SR | SR | Sub-SR | SR | Sub-SR | SR | Sub-SR | SR | Sub-SR | SR | Sub-SR | SR | Sub-SR |
| 3D-LLM (Finetuned) | 10.4 | 20.3 | 9.1 | 18.5 | - | - | - | - | - | - | - | - | - | - | - | - |
| Everything in Context | 35.5 | 63.9 | 32.4 | 45.2 | - | - | - | - | - | - | - | - | - | - | - | - |
| Most Recent Memory | 32.8 | 62.3 | 23.4 | 38.6 | 20.1 | 34.8 | 12.4 | 25.3 | 10.4 | 20.7 | 5.4 | 12.1 | 21.1 | 39.3 | 13.7 | 25.3 |
| Retrieval-Augmented Memory | 34.2 | 63.0 | 28.3 | 46.2 | 21.8 | 40.2 | 13.7 | 28.0 | 10.8 | 21.6 | 4.8 | 10.6 | 22.3 | 41.6 | 15.6 | 28.3 |
| 3DLLM-MEM (Ours) | 45.5 | 73.4 | 37.0 | 65.4 | 36.8 | 67.8 | 31.6 | 57.4 | 30.5 | 46.2 | 27.8 | 42.1 | 37.6 | 62.5 | 32.1 | 55.0 |

(a) Results on 3DMEM-BENCH embodied tasks. SR stands for success rate. Sub-SR stands for sub-success rate. Our model outperforms existing approaches by a large margin.

| Model | Embodied Task | | Embodied Question Answering (EQA) | | | | | Captioning | | |
|---|---|---|---|---|---|---|---|---|---|---|
| | In-domain | In-the-wild | Spatial | Nav. | Comparative | Layout | Count | BLEU1 | BLEU4 | METEOR |
| 3D-LLM (Finetuned) | - | - | 2.9 | 5.8 | 0.0 | 7.7 | 0.0 | 42.3 | 12.0 | 30.6 |
| 3D-Mem (GPT4-o) | - | - | 39.9 | 11.0 | 25.8 | 19.1 | 7.8 | 41.7 | 4.7 | 31.8 |
| 3D-Mem (Gemini-2.5-Flash) | - | - | 41.6 | 18.2 | 37.6 | 30.2 | 12.7 | 42.8 | 4.8 | 29.6 |
| 3D-Mem (Gemini-2.5-Pro) | - | - | 39.7 | 27.7 | 36.0 | 35.2 | 16.4 | 41.5 | 3.0 | 28.6 |
| Most Recent Memory | 21.1 | 13.7 | 27.5 | 30.2 | 24.3 | 20.1 | 10.5 | 32.4 | 10.1 | 25.6 |
| Retrieval-Augmented Memory | 22.3 | 15.6 | 38.0 | 33.4 | 31.8 | 29.7 | 15.6 | 40.8 | 11.5 | 29.3 |
| 3DLLM-MEM (Ours) | 37.6 | 32.1 | 62.8 | 40.6 | 41.4 | 39.9 | 26.3 | 58.2 | 18.8 | 37.3 |

(b) Results on all tasks in 3DMEM-BENCH. Average success rate is reported for embodied tasks. *Nav.* stands for long-term object navigation. We report accuracy score for open-ended EQA evaluation and follow the standard LLM-as-judge evaluation protocol by prompting Gemini. Evaluation details are provided in Appendix E.

Table 2: Comparison with 3D memory models and standard memory management approaches. Our model, 3DLLM-MEM, achieves the best performance across embodied, EQA and captioning tasks.

3DLLM-MEM in §4.2, along with qualitative results. Finally, in §4.3, we conduct an ablation study of key design choices in 3DLLM-MEM, demonstrating the effectiveness of our proposed memory fusion mechanism.

## 4.1 Experimental Setup

**Implementation details** We implement our model based on LLaVA-3D (Zhu et al., 2024), modifying it to be compatible with Google TPUs with PyTorch/XLA frameworks (Paszke et al., 2019; team, 2017–2025) . We first expand the model's context window to 8192 tokens to accommodate long-term memory inputs. We then fine-tune our proposed memory module along with the LLM decoder using our training split, initializing from LLaVA-3D's pretrained weights. Training is conducted on 8 Google Cloud TPU v5p cores with a batch size of 256. Our model is trained using supervised fine-tuning (SFT) with a standard language modeling loss. More details are provided in Appendix D.

**Baselines** We compare 3DLLM-MEM against a broad range of memory management approaches:
- **Everything in Context.** For a small subset of scenes, it is feasible to fit all observations directly into the model's context window.
- **Most Recent Memory.** Since retaining all observations in context is infeasible, we keep only the most recent observations, assuming they are most relevant to the current task.
- **Retrieval-Augmented Memory.** Inspired by retrieval-based techniques, we adopt a memory bank that stores past observations. During inference, the most relevant memory entries are retrieved and appended before the working memory to augment reasoning.
- **3D-LLM** (Hong et al., 2023b). A popular 3D LLM recognized by the community. We finetune it on our training data and report its performance using the "everything in context" strategy with the longest context window supported. Further details are provided in Appendix G.
- **3D-Mem** (Yang et al., 2025b). A framework designed for 3D scene memory in embodied exploration and reasoning. However, this method does not support embodied interaction or action execution.

## 4.2 Experimental Results

**Results on embodied tasks** As shown in Table 2a, 3DLLM-MEM significantly outperforms all existing approaches on both in-domain and in-the-wild embodied tasks. Notably, while the performance of other methods drops sharply in the in-the-wild setting, our method demonstrates strong generalization capabilities with a average success rate of 32.1%. 3D-LLM showcases the lowest

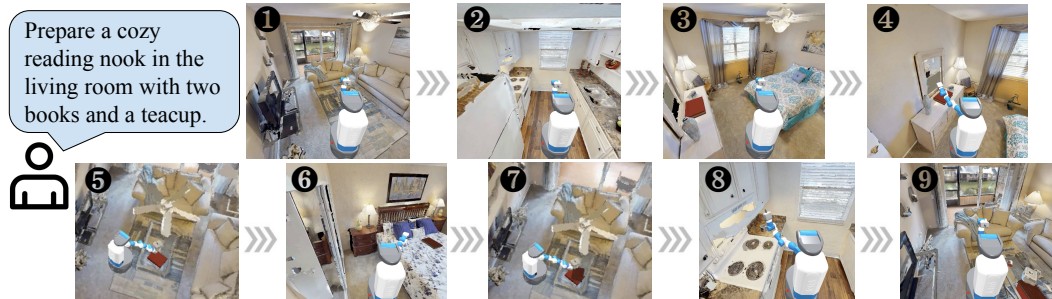

Figure 4: Qualitative example of 3DLLM-MEM, which maintains and utilizes a long-term memory to complete the task. Detailed task execution trajectory can be found in Figure 6.

performance even under simple task settings, highlighting the necessity of incorporating an explicit memory module. Both the *Most Recent Memory* and *Retrieval-Augmented Memory* (RAG) baselines perform poorly in this setting, with RAG showing only a slight improvement, highlighting the challenges of retrieving relevant episodic memory. Interestingly, the *Everything in Context* baseline performs better than both recent memory and RAG approaches, suggesting that when all information can fit within the context window, the model can effectively utilize it. However, 3DLLM-MEM still outperforms *Everything in Context*, indicating the benefits of selectively fusing task-relevant memory features to better guide embodied reasoning and execution. As task complexity increases from simple to hard, all existing approaches degrade significantly, achieving only ∼5% success rate in hard in-the-wild tasks. In contrast, 3DLLM-MEM maintains a strong performance of 27.8%, demonstrating its scalability and effectiveness in managing longer-term memory representations.

**Results on long-term EQA and captioning**   As shown in Table 2b, 3DLLM-MEM consistently outperforms all existing approaches across all tasks in our benchmark. Notably, 3D-LLM achieves the second-best performance on the captioning task, highlighting its strong ability to summarize object-centric semantic memory. However, due to limited context length, it performs poorly on the EQA tasks, which require long-term spatial-temporal reasoning. 3D-Mem demonstrates improved performance in EQA over other baseline approaches. However, it falls short on spatial relation, navigation and object counting tasks, indicating the limitation of relying solely on aggregated image-centric memories. 3DLLM-MEM significantly outperforms both *Most Recent Memory* and *RAG Memory*, which further demonstrates the effectiveness of our memory fusion technique.

**Qualitative results**   We provide qualitative examples in Figure 4 and a more detailed version with explanations in Figure 6 (Appendix H), demonstrating that 3DLLM-MEM is capable of maintaining long-term memory and executing complex tasks in embodied environments.

### 4.3   Ablation Study

Our approach initializes the fused memory using working memory features, aiming to fuse the most relevant memories for the current task. We ablate several design choices for initializing the fusion query, as shown in Table 3. When using either the most recent episodic memory or learnable zero parameters, performance degrades compared to our proposed method. Interestingly, using the most recent memory outperforms zero initialization in the simple setting but underperforms in the hard setting. One possible explanation is that recent memory initialization encourages fusion with nearby observations, which may be sufficient for simple tasks and leads to faster convergence. In contrast, zero initialization is guided solely by training supervision to learn which memories are most useful. In summary, the ablation results demonstrate that initializing fusion queries with working memory tokens provides the most effective and robust design choice for long-term memory fusion.

## 5   Related Works

**3D Large Language Models**   3D Large Language Models (3D-LLMs) have demonstrated promising results across a wide variety of tasks, including 3D scene understanding, object detection, and segmentation (Hong et al., 2023b; Zhou et al., 2024; Huang et al., 2024a; Chen et al., 2024b; Xu et al., 2025a). In parallel, 3D embodied agents have expanded these capabilities to planning and action in interactive environments (Brohan et al., 2023; Huang et al., 2024b; Chen et al., 2024a; Black et al.,

| Model | Simple | | | | Medium | | | | Hard | | | | Average | | | |
|---|---|---|---|---|---|---|---|---|---|---|---|---|---|---|---|---|
| | In-domain | | In-the-wild | | In-domain | | In-the-wild | | In-domain | | In-the-wild | | In-domain | | In-the-wild | |
| | SR | Sub-SR | SR | Sub-SR | SR | Sub-SR | SR | Sub-SR | SR | Sub-SR | SR | Sub-SR | SR | Sub-SR | SR | Sub-SR |
| 3DLLM-MEM | 45.5 | 73.4 | 37.0 | 65.4 | 36.8 | 67.8 | 31.6 | 57.4 | 30.5 | 46.2 | 27.8 | 42.1 | 37.6 | 62.5 | 32.1 | 55.0 |
| Init with Most Recent Episodic Memory | 42.3 | 69.4 | 28.6 | 50.7 | 32.4 | 58.6 | 23.7 | 45.1 | 22.6 | 37.8 | 15.3 | 31.4 | 32.4 | 55.3 | 22.5 | 42.4 |
| Init with Learnable Zero Parameters | 41.4 | 67.2 | 27.9 | 50.0 | 33.0 | 59.2 | 23.4 | 45.8 | 24.2 | 40.4 | 18.6 | 35.6 | 32.9 | 55.6 | 23.3 | 43.8 |

Table 3: Ablation study of query initialization designs in our memory fusion module.

2024). Yet, existing models face significant challenges when performing long-horizon embodied tasks in densely populated 3D environments that require reasoning over long-term spatial-temporal memory. To address this, we propose an explicit memory module inspired by the structure of human implicit and explicit memory. Our model employs a memory fusion mechanism that efficiently retrieves and learns task-relevant information, resulting in enhanced performance on complex embodied tasks.

**Long-term Embodied Trajectories**  Embodied AI simulators (Chang et al., 2017; Kolve et al., 2017; Szot et al., 2021; Shen et al., 2021) have fostered the development of embodied AI agents. Grounded in these environments, some existing benchmarks focus on high-level planning tasks, typically involving short trajectories that can often be completed within single-room settings, thereby requiring minimal spatial-temporal memory (Shridhar et al., 2020, 2021; Li et al., 2024a; Szot et al., 2024; Li et al., 2024b; Yang et al., 2025a). Other benchmarks emphasize long-term scene exploration with extended trajectories, but are primarily centered around navigation tasks and often lack embodied interaction support (Deitke et al., 2020; Ramakrishnan et al., 2021; Krantz et al., 2022; Khanna et al., 2024). To bridge this gap, we introduce 3DMEM-BENCH, a benchmark specifically designed to evaluate long-horizon task execution that requires rich spatial-temporal memory and full embodied task support, as summarized in Table 1.

**Embodied Question Answering Benchmark**  Embodied Question Answering (EQA) benchmarks (Das et al., 2018; Wijmans et al., 2019; Yu et al., 2019) have been developed to advance goal-driven agents that can perceive their environment. Some EQA benchmarks also include embodied memory QA evaluation, such as OpenEQA (Majumdar et al., 2024), which includes an episodic memory QA split, and Yang et al. (2024), which focuses on spatial memory QA. In contrast, our benchmark, 3DMEM-BENCH jointly targets both spatial and episodic memory, especially their changes over time, while also supporting embodied action tasks, EQA and captioning. For specific comparison on EQA, our long-term memory EQA tasks are designed to require reasoning over multiple "pieces" of memory and their changes across time and space. Additionally, we consider the agent's location in the scene at the moment of answering each question during evaluation.

**Memory System**  Memory is a fundamental component of AI systems, with early work in the context of LLM agents that utilize memory for decision-making in web-based and sandbox environments (Shinn et al., 2023; Zhang et al., 2023; Packer et al., 2023; Zhang et al., 2024). Most existing approaches construct an experience pool or memory bank and focus on improving the retrieval of useful past information (Zhao et al., 2024; Gao et al., 2024; Xu et al., 2025b). In the computer vision domain, temporal memory has been studied extensively in video understanding and generation tasks (Wang et al., 2021; Diao et al., 2025), while spatial memory has been applied to scene-level visual understanding and 3D reconstruction (Wang and Agapito, 2024; Zou et al., 2025). Recent work such as 3D-Mem (Yang et al., 2025b) has investigated 3D scene memory for exploration and reasoning by prompting vision-language models. In contrast, our work focuses on dense 3D memory representations that are critical for real-world embodied scenarios, where task execution depends heavily on maintaining and reasoning over long-term spatial-temporal memory.

# 6  Conclusion

In this work, we introduce 3DMEM-BENCH, a comprehensive benchmark containing fine-grained long-term memory embodied tasks—ranging from simple to hard—along with question-answering tasks that target memory changes across time and space, and captioning task in complex 3D environments. We propose 3DLLM-MEM, an embodied 3D-LLM with novel memory fusion approach for spatial-temporal reasoning, planning, and acting. One limitation of our model is that currently 3DLLM-MEM does not involve low-level navigation and control policy, but utilizes high-level pre-

defined policies in simulator for carrying out the actions. We think that such aspects are orthogonal to our study, and could be explored and seamlessly integrated into our framework in the future.

## Acknowledgments and Disclosure of Funding

We thank anonymous reviewers and other members of UCLA-NLP+ group for their helpful comments. This work was partially supported by U.S. DARPA ECOLE Program No. HR00112390060, ONR grant N00014-23-1-2780, Amazon Research Award, and a Google gift fund. Peng and Chang have financial COI with Google and Amazon and were supported in part by a grant from DARPA to the Simons Institute for the Theory of Computing.

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

## A  Broader Impact

The deployment and release of 3DLLM-MEM carry both potential benefits and risks. These considerations include visual aspects as well as common issues found in existing LLMs like Alpaca and Vicuna. Since 3DLLM-MEM is built on LLaMA, Vicuna, and CLIP, it inherits certain challenges associated with LLMs and vision encoders. Below, we outline the risks and the mitigation strategies implemented for the release of this model.

**Hallucination**   Similar to other LLMs, 3DLLM-MEM might produce outputs that are not based on factual information or input data. This raises concerns about the accuracy of inferences, particularly in critical applications such as medical fields.

**Biases**   Biases present in the base models can be brought to 3DLLM-MEM, stemming from both the vision encoder (CLIP) and the language decoder (LLaMA / Vicuna). This may result in biased outcomes or unfair representations of diverse content.

**Energy Consumption**   We train our model on our training data split which contains about 26K trajectories. The training time only takes less than one day, which makes energy consumption not a primary concern.

## B  Environment Construction

To support navigation-centric interaction, the agent requires precise knowledge of two things: the traversable layout of each scene and the exact locations of all movable objects. Following 3D-CLR (Hong et al., 2023a), we build this spatial substrate from HM3D's richly annotated indoor scans. We rely on the semantic surface mesh that accompanies each scene to calculate the room and objects' locations. The mesh scan has a unique (24-bit) hexadecimal color for every surface triangle that provides a semantic table that links each color to a surface label (e.g., floor or ceiling) and a room identifier.

We first derive an axis-aligned bounding box for every room in each HM3D scene. We query the semantic table to retrieve, for every room identifier, the hex colors assigned to its floor and ceiling surfaces. Before processing individual rooms we aggregate the minimum heights of all floor surfaces in the entire scan, producing a global set of candidate floor elevations. For each room we then load the point clouds of its floor and ceiling. When both are available, the vertical bounds are given by the floor's lowest point and the ceiling's highest point. If the floor is missing, the lower bound is set to the highest global floor elevation that still lies below the room's ceiling; if the ceiling is missing, the upper bound defaults to the highest point in the available cloud (i.e., the surface immediately above the floor). Rooms for which both surfaces are absent are discarded because no reliable vertical evidence is present. Horizontal limits are simply the minimum and maximum coordinates of the

room's floor and ceiling points, and together these six coordinates constitute the room's axis-aligned bounding box.

Based on the bounding boxes for each room, we calculate the bounding boxes of objects in that room by gathering the corresponding vertices and computing the bounding boxes of the object by taking the minimum and maximum coordinates along the three axes. We merge room-level and object-level bounding boxes by their shared room index. After the aforementioned processing, the dataset comprises 182 distinct 3-D scenes covering a total of 2,602 rooms.

## C Embodied Task Data Examples

In this section, we provide complete embodied task trajectories.

Here is an example of in-the-wild task:
**Task**: Rearrange desk items by size. The items to be rearranged are: vase, box, and printer.
**Trajectory**:
"I'm in room 10. I will start exploring the room for the items I need.", "I see a printer(0) on the desk(0). I also see a box(0) on the cabinet(0).", "I need to remember their locations and sizes for comparison later.",
"<GO TO NEW ROOM>",
"Now I'm in room 11. I'm exploring this room for the items.", "I don't see a vase in the room, but I see a printer(0) on a rack(1).", "I will remember these objects and their locations.",
"<GO TO NEW ROOM>",
"Now I'm in room 6. I'm exploring this room.", "I do not see any of the items I am looking for.",
"<GO TO NEW ROOM>",
"I'm in room 8. I am exploring this room.", "I see flower vase(0) on the floor. I will remember the object and its location.",
"<GO TO NEW ROOM>",
"I'm in room 7. I am exploring this room.", "I do not see any of the items I am looking for.",
"<GO TO NEW ROOM>",
"I'm in room 9. I am exploring this room.", "I do not see any of the items I am looking for.",
"<GO TO NEW ROOM>",
"Now I am in room 5. I am exploring this room.", "I do not see any of the items I am looking for.",
"<GO TO NEW ROOM>",
"Now I am in room 2. I am exploring this room.", "I see a box(0) on the armchair(1). I will remember these objects and their locations.",
"<GO TO NEW ROOM>",
"I'm in room 1. I do not see any items that are part of the rearrangement.",
"<GO TO NEW ROOM>",
"Now I'm in room 12. I will start exploring this room.", "I think I've seen the vase, printer, and box in other rooms, so I'll head back to room 8 to pick up the vase first.",
"<GO TO ROOM(8)>",
"I need to pick up flower vase(0) in this room first. I pick up flower vase(0).",
"<PICK UP flower vase(0) from room(8) in room(8)>",
"I'm carrying flower vase(0) to room 10 where I saw printer and box.",
"<GO TO ROOM(10)>",
"Now I am in room 10. I will put flower vase(0) on the desk(0) for now so I can pick up box(0).",
"<PUT DOWN flower vase(0) from room(8) on desk(0) in room(10)>",
"Now I will pick up box(0) from cabinet(0).",
"<PICK UP box(0) from room(10) in room(10)>",
"Now I will put box(0) on the desk(0) so I can pick up printer(0).",
"<PUT DOWN box(0) from room(10) on desk(0) in room(10)>",
"Now I will pick up printer(0) from beside the desk(0).",
"<PICK UP printer(0) from room(10) in room(10)>",
"I need to compare the sizes of the vase, box and printer now to determine the rearranging order.",
"I recall that printer(0) in this room had the largest bounding box dimensions based on my earlier observation. I need to put down printer(0).",
"<PUT DOWN printer(0) from room(10) on desk(0) in room(10)>",
"Now I will pick up box(0).",

"<PICK UP box(0) from room(10) in room(10)>",
"I remember that box(0) from cabinet(0) had medium bounding box dimensions, so I will put it on the desk.",
"<PUT DOWN box(0) from room(10) on desk(0) in room(10)>",
"Now I will pick up flower vase(0).",
"<PICK UP flower vase(0) from room(8) in room(10)>",
"I remember that flower vase(0) had the smallest bounding box dimensions, so I will put it on the desk.",
"<PUT DOWN flower vase(0) from room(8) on desk(0) in room(10)>",
"I will now arrange the objects from small to large: flower vase(0), box(0) and printer(0) on the desk(0).", "Now I pick up the medium object, box(0).",
"<PICK UP box(0) from room(10) in room(10)>",
"I will put the medium object on the left of the largest object, printer(0).",
"<PUT DOWN box(0) from room(10) on desk(0) in room(10)>",
"Now I pick up the smallest object, flower vase(0).",
"<PICK UP flower vase(0) from room(8) in room(10)>",
"I will put the smallest object on the left of the medium object, box(0).",
"<PUT DOWN flower vase(0) from room(8) on desk(0) in room(10)>",
"The objects are now arranged from small to large: flower vase(0), box(0) and printer(0) on the desk(0). Task Complete."

Here is an example of in-domain task:

**Task**: Preparing for Cooking Stir-fried Tomato and Scrambled Eggs. You need to prepare the ingredients, seasonings and cooking tools.

**Trajectory**: "I'm at room 4. It seems to be living room, I need to prepare ingredients and seasonings and cooking tools and setup them in the kitchen", "There is seasonings on the table.",
"<PICK UP seasonings(0) from room(4) in room(4)>",
"No other things found for my task",
"<GO TO NEW ROOM>",
"I'm at room 5. It seems to be dining room. I saw an apron on dining table.", "I need the apron but I have seasonings(0) on my hand. Lets find kitchen to put the seasonings first and go back for apron",
"<GO TO NEW ROOM>",
"I'm at room 8. It seems to be kitchen with two cabinets. I found tomatoes and eggs on the cabinet(1) that is far away from me and cooking pan in the sink.", "There are two countertops in this room. countertop(1) is closer to stove(0). I should setup on countertop(1).",
"<PUT DOWN seasonings(0) from room(4) on countertop(1) in room(8)>",
"<PICK UP tomatoes(0) from room(8) in room(8)>",
"<PUT DOWN tomatoes(0) from room(8) on countertop(1) in room(8)>",
"<PICK UP eggs(0) from room(8) in room(8)>",
"<PUT DOWN eggs(0) from room(8) on countertop(1) in room(8)>",
"<PICK UP cooking pan(0) from room(8) in room(8)>",
"<PUT DOWN cooking pan(0) from room(8) on stove(0) in room(8)>",
"I remember I saw apron in room 5. I need to find it back.",
"<GO TO ROOM(5)>",
"<PICK UP apron(0) from room(5) in room(5)>",
"<GO TO ROOM(8)>",
"<PUT DOWN apron(0) from room(5) on countertop(1) in room(8)>",
"The setup in kitchen has been prepared. Task Complete."

# D   Implementation Details

We implement our model based on LLaVA-3D (Zhu et al., 2024), modifying it to be compatible with Google TPUs with PyTorch/XLA frameworks (Paszke et al., 2019; team, 2017–2025) . We first expand the model's context window to 8192 tokens to accommodate long-term memory inputs. We then fine-tune our proposed memory module along with the LLM decoder using our training split, initializing from LLaVA-3D's pretrained weights. Training is conducted on 8 Google Cloud TPU v5p cores with a batch size of 256 for 1000 steps, which takes about 1 day to complete. We use Adam optimizer with learning rate of 2e-5 with no weight decay. Additionally, we apply a linear warmup

of the learning rate during the initial 3% steps, increasing from $10^{-8}$ to $10^{-5}$, followed by a cosine decay scheduler.

## E  Prompts for Gemini

As mentioned in § 2.2, we prompt Gemini to generate the long-term trajectories as illustrated in Table 4, generate the question-answering tasks as shown in Table 5, and generate caption tasks as shown in Table 6. For open-ended QA evaluation, we followed standard LLM-as-judge protocol by prompting Gemini as illustrated in Table 7.

## F  Data Validation

### F.1  Trajectory Validation

We implement a trajectory simulation pipeline driven by the commands listed in Table 4. For each command, the simulator records the agent's current room and the full set of objects it is holding, then updates the set of objects in each room to reflect pick-up and put-down actions. A pick-up removes the specified object (along with any nested items) from the room the agent occupies and adds it to the agent's hand; a put-down removes the object from the agent's hand and places it into the designated room. The pipeline validates each command based on these criteria: (1) the agent's location; (2) the referenced object and (3) the correctness of pick-up and put-down actions. For location validation, a command is marked as invalid if the agent attempts to pick up an object from a room that does not match its current room, or tries to drop an object into a room other than the one it currently occupies. Additionally, if the agent tries to visit a room that does not exist in the scene, or attempts to enter a new room when all rooms have already been explored, the trajectory is also considered invalid. For object validation, a pick-up command is invalid if the target object does not exist in the current room, and a put-down command is invalid if the agent is not currently holding the specified object. For pick-up and put-down validation, the agent is allowed to hold only one object at a time. A command is considered invalid if the agent attempts to pick up an object while already holding one, or tries to put down an object when its hand is empty. Finally, after all commands have been executed, if the trajectory ends with the agent still holding an object that was never put down, the entire trajectory is marked as invalid.

### F.2  Human Validation

As mentioned in §2.3 After automatic trajectory validation, we further conduct human validation, in which four student experts in the field manually inspect each benchmark example. We render multi-view images of the entire scene using the simulator and verify whether the benchmark annotations accurately correspond to the simulated environment as illustrated in Figure 5.

## G  Evaluation Setup Details

**3D-LLM**  Similar to the 3D-LLM work (Hong et al., 2023b), we use their direct reconstruction method to extract the 3D features from each scene in our training data. To process our long-term memory data, which requires multi-scene input across each task, we feed each room in the task through the 3D-LLm Q-Former head independently to get separate 32-token dense representation of each room with per-room 3d positional embeddings injected into the features. Then we concatenate the representations before feeding the input into the frozen t5-flanxl (Chung et al., 2022) backbone like the original work.

The 3D-LLM model also included learned location tokens used to describe certain locations within each room in the scene. To fit 3D-LLM to our task data, we substitute the location tokens with our specific interaction tokens (eg. <GO TO ROOM> used by all models in our experiments) and train the model to learn the new tokens to stay consistent with our higher level interaction used across our training data. Analysis of the 3D-LLM model evaluation output, indicated the primary struggle for the model was retaining long term memory of semantic observations in the scene, so we prioritized aligning 3D-LLM with the high level long-term memory representation in our data over low level spatial understanding of the scene.

Table 4: Prompt template for generating task trajectories. {In-context examples} are in-context examples. {Input scene information} are scene, room and object semantics along with their bounding boxes.

Our longer task data input also required truncation to fit within the 512 token context length of 3D-LLM's t5-flanxl backbone. We retain the task description and move the question to the beginning of the prompt for the QA data to ensure the model still receives the information necessary to understand its tasks. The longer trajectory of past events is then the only information that gets truncated before fed into the t5 encoder.

> **Prompt**
>
> You are an AI assistant / task generator in the room. All object instances in this 3D scene are given, along with their bounding boxes and ids." Each object's bounding boxes are represented by a 3D coordinate '<obj_name>(num)': [x min, y min, z min],[x max, y max, z max]' with units of meters, and each represents left-bottom corner and the right-top corner coordinate.
>
> You will also receive a trajectory composed of the following tokens and reasoning chains. <GO TO ROOM(id)>: which navigates back to a specific room (id). This can only be done if the agent already go to this room. <PICK UP object_name(id) from room(id) in room(id)>: Pick up an object that originally belongs to a specific room while in that same room. <PUT DOWN object_name(id) from room(id) on object_name(id) in room(id)>: Place an object (that originally belongs to a room) onto another object (such as a table or floor) in a room. <GO TO NEW ROOM>: which navigates to a new room you haven't explored and unlocks objects there.
>
> This trajectory is what the agent have executed over the past. You need to propose several questions and answers that focused on the reasoning abilities of the long-term memory of the agent. These reasoning questions should focus on what have changed temporally or spatially in this agent's memory. It's important that this change challenged the agent's memory. For example the questions should contain object counting, spatial relation, comparison between objects across rooms, long-term multi-room room layout, long-term multi-room object navigation. Remember spatial memory is important, you should design questions that asked about the 3D object spatial relation and layout in the room that need the agent to perform a hard reasoning for the final answer.
>
> For clarity, consider these examples: {In-context examples}
> ―――――――――
> Here is the scene information: {Input scene information}
> Here is the agent's trajectory: {Input agent's trajectory}

Table 5: Prompt template for generate QA data. {In-context examples} are in-context examples. {Input scene information} are scene, room and object semantics along with their bounding boxes. {Input agent's trajectory} is the 3D agent's explored trajectories and action chains.

For finetuning on our data, we use the hyperparameters provided by 3D-LLM and finetune until model loss stops decreasing. Due to compute limitations, we trained on captioning task for 15 epochs, question-answering task for 20 epochs, and allocated most of the compute time on the embodied task, which we trained on for 75 epochs.

**3D-Mem**  We benchmark 3D-Mem (Yang et al., 2025b) on the question-answering and captioning splits of 3DMEM-BENCH. 3D-Mem is a snapshot-based 3D memory architecture originally developed for embodied exploration and reasoning; it keeps two complementary stores—memory snapshots, a compact set of multi-view RGB-D frames with per-object bounding boxes summarizing the areas the agent has inspected, and frontier snapshots, boundary views that suggest where useful new information may be found next. In its native setting the agent navigates an unfamiliar scene by selecting the frontier view most likely to advance its task and then answers visual questions using the most relevant memory snapshots. Because our evaluation focuses on post-exploration reasoning rather than active exploration, we disable the frontier component and retain only the memory snapshots. For these two tasks, the system will capture memory snapshots in each room from the room center, and finish the QA and captioning base on the memory snapshots of all the explored rooms.

# H  Qualitative Examples

We provide qualitative examples as shown in Figure 6. It demonstrates that 3DLLM-MEM can maintain a long-term memory and perform complex tasks in the embodied environments. More examples can be found in the **supplementary materials**.

**Prompt**

You are provided with a scene description containing multiple rooms. Each room includes a list of objects along with their positions in the room, represented by bounding boxes. Each object's bounding box is defined by a 3D coordinate in the format: <object_name>(num): [x min, y min, z min],[x max, y max, z max] with units in meters (defining the left-bottom and right-top corners). Your task is to generate an object caption for each room in the form of a coherent, descriptive paragraph that conveys the 3D spatial arrangement and relative positions of all objects within that room.

Then, you will receive the object descriptions and caption for the current 3D room you are in. You will also be provided with the previous rooms' captions as well. Your task is to generate new captions covering the summarization of the common features across all rooms based on your current room and important difference based on your current room. The reasons of generating the new caption is to help the agent to remind of what are in previous rooms memories can help the agent in this current room. The past objects and observations should be related to current room by examining the summarization of common things and differences. For clarity, consider these examples: {In-context examples}
———————————
Here is the scene information: {Input scene information}
Here is current room you are in and previous rooms you went: {Input agent's location}

Table 6: Prompt template for generate QA data. {In-context examples} are in-context examples. {Input scene information} are scene, room and object semantics along with their bounding boxes. {Input agent's location} is the location for current room in the scene and the past explored rooms.

**System message**

Please act as an impartial judge and evaluate the quality of the response provided by an AI assistant to the user question. Your evaluation should consider correctness and helpfulness. You will be given a reference answer and the assistant's answer. You evaluation should focus on the assistant's answer to the second question. Begin your evaluation by comparing the assistant's answer with the reference answer. Identify and correct any mistakes. Be as objective as possible. After providing your explanation, you must rate the response on a scale of 1 to 10 by strictly following this format: "[[rating]]", for example: "Rating: [[5]]".

**Prompt**

<|The Start of Reference Answer|>
### User:
question_1
### Reference answer:
ref_answer_1
### User:
question_2
### Reference answer:
ref_answer_2
<|The End of Reference Answer|>
<|The Start of Assistant A's Conversation with User|>
### User:
question_1
### Assistant A:
answer_1
### User:
question_2
### Assistant A:
answer_2
<|The End of Assistant A's Conversation with User|>

Table 7: Prompt template for open-ended QA evaluation following standard LLM-as-judge protocol.

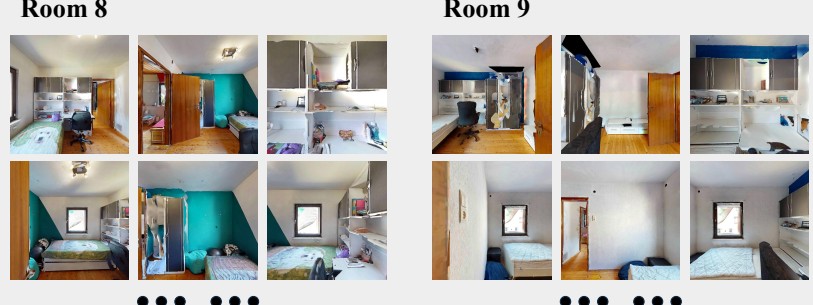

**Room 8**          **Room 9**

● ● ●  ● ● ●          ● ● ●  ● ● ●

## Object Captioning

Room 8 contains distinct sleeping, working, and lounging zones. **Bed**(0) extends along one side, with **pillow**(0) near its head and **cabinet**(1) beside it, which supports **decoration**(0) and is flanked by **book**(0) and **book**(1). Near the foot of bed(0), **yoga mat**(0), **bag**(0), and **box**(0) form a small activity area. At the far end, **wardrobe**(0) stands near wall(4), next to **beanbag chair**(0) and **exercise ball**(0), with **clothes**(0) draped nearby, creating a relaxed corner. The office area, positioned across from bed(0), includes **desk**(0), **office chair**(0), and **cabinet**(0), with **trashcan**(0) nearby and **ceiling lamp**(0) overhead. **Window**(0) on the **parapet**(0) brings in natural light. The arrangement ensures functional use of space while maintaining clear separation between different activities.

## Question Answering

### Object Counting
- Q: How many chairs are there in Room 8?
- A: There are 2 chairs in total: one office chair(0) and one beanbag chair(0).

### Spatial Relation
- Q: If you are sitting on the office chair(0) working at the desk(0) in Room 8, where is the window(0) relative to you?
- A: The window(0) is behind you to the left.

### Comparative QA
- Q: Which room has more books, Room 8 or Room 9?
- A: Room 8 has more books. It contains two books, book(0) and book(1), both placed on top of the cabinet(0) near the desk(0).

### Object Navigation
- Q: If you are sitting on the beanbag chair(0) in Room 8, how do you get to the books?
- A: You need to stand up, turn left, and walk across the room toward the cabinet(0) near the desk(0), where the books are placed.

### Room Layout
- Q: In Room 8, what is the relative layout of the bed(0), desk(0), and office chair(0)? Which one is closest to the window(0)?
- A: The bed(0) is placed along the side wall near the window(0), the desk(0) is located in the far corner of the room along wall(5), and the office chair(0) is positioned directly in front of the desk(0). Among these, the bed(0) is closest to the window(0), which is embedded in the wall beside it.

Figure 5: Example of human annotators manually check the data quality on QA and captioning tasks through multiple rendered multi-view images from each room.

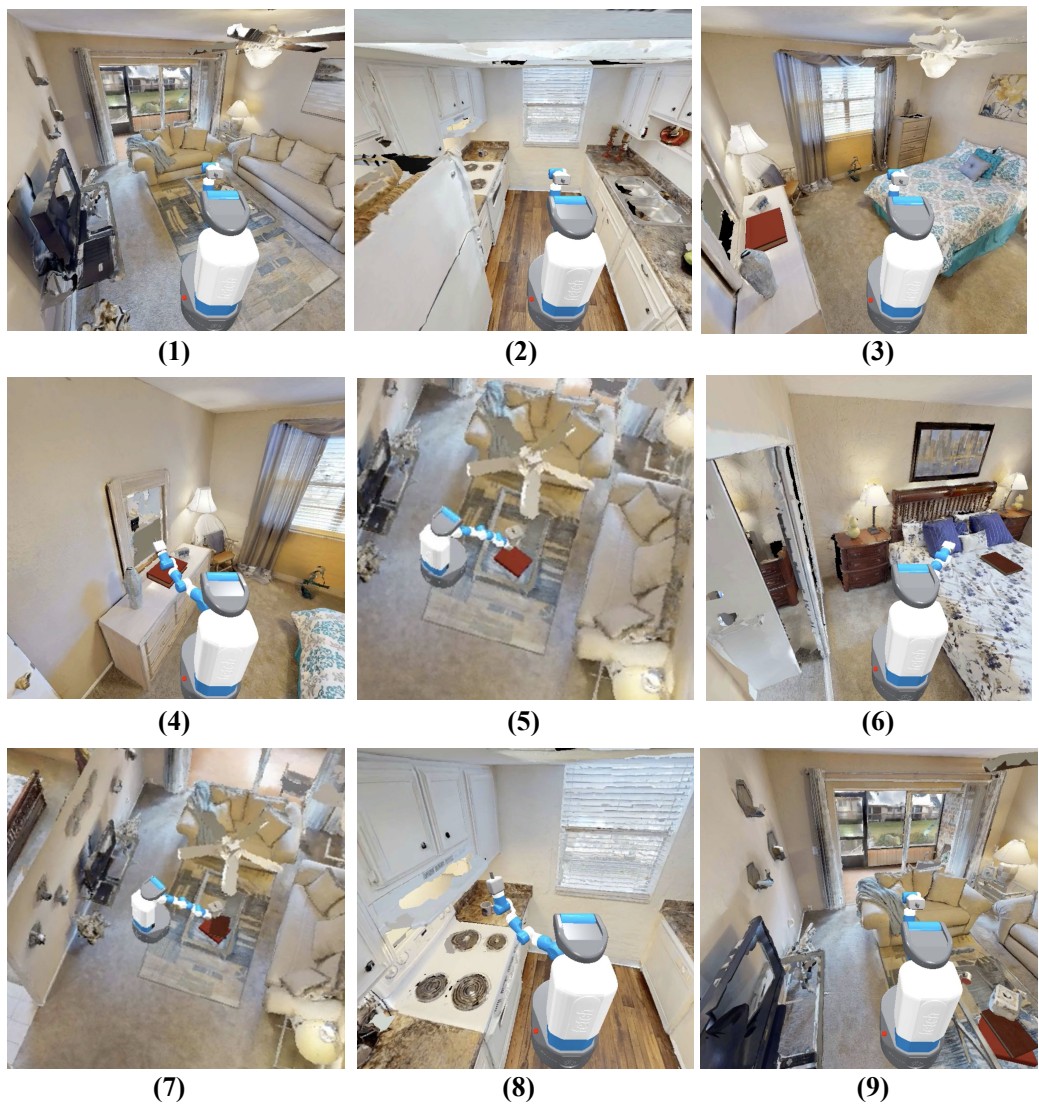

Figure 6: Qualitative example of 3DLLM-MEM. The task instruction is: *Prepare a cozy reading nook in the living room with two books and a teacup*. In images (1) and (2), the agent explores the environment randomly, forming an initial memory of the scene. After receiving the task instruction, it recalls its memory and navigates to the bedroom to pick up a book from the cabinet, as shown in images (3) and (4). The agent then returns to the living room and places the book on the table in front of the sofa (image 5). Unable to recall any additional books, the agent resumes exploration and finds a second book on the bed, which it picks up (image 6) and stacks on top of the first book (image 7). Finally, the agent recalls seeing a teacup in the kitchen, navigates to retrieve it (image 8), and places it on the table in the living room (image 9). The task is successfully completed.

