# OpenReview forum: "3DLLM-Mem: Long-Term Spatial-Temporal Memory for Embodied 3D Large Language Model"
_NeurIPS.cc/2025/Conference — NeurIPS 2025 poster_

### Official Review · Reviewer_kteJ · 2025-06-11

**Clarity:** 3
**Significance:** 2
**Originality:** 3
**Rating:** 4
**Confidence:** 3

**Summary:**

This paper first proposes a 3D benchmark that is called 3DMEM-Bench, covering 26000 trajectories and 2892 embodied tasks. Then this paper proposes the formulation of 3DLLM-EME, which is a model with dyanmic memory management and fusion for embodied spatial-temporal reasoning and actions. In details, this model us working memory tokens to select from episodic memory. Experiments on 3DLLM-MEM shows the effect of the proposed method.

**Questions:**

1. What is the efficiency of the proposed model compared with other methods?

2. How to adopt this model into scenes with more dynamics and interaction?

3. What is the contribution of each component in your framework?

**Ethical Concerns:**

["Major Concern: Safety and security"]

**Final Justification:**

After reading the paper, rebuttal from authors, and the comments from other reviewers, I think this paper has some significant contributions, such as building a new 3D benchmark and proposing the working and episodic memory. However, I think some concepts are not convincing, such as I still do not think this work can solve the dynamic scenes. Thus, I give the borderline score, but prefer to accept.

**Limitations:**

No limitation is claimed in this paper. I think this paper should discuss more about their limitations in dealing with various scenes.

**Paper Formatting Concerns:**

N/A.

**Quality:**

2

**Strengths And Weaknesses:**

Strength:

This paper builds a new 3D benchmark, and the idea of working memory and episodic memory is interesting.

Weakness:

1. I think the idea of querying tokens from the episodic memory is rational. However, it may cause the huge computation for reasoning. The authors should list the efficiency comparison. Moreover, if the thinking process is too long, the environment will be changed a lot, and the past observations and interactions might be useless.

2. Especially, I think this model can be only adopted to the static environment, while cannot handle the dynamic environment. Especially, the interactions with others are also not described.

3. This model is built on the base model of LLaVA-3D. Thus, I wonder whether the advantage of 3DLLM-MEM is derived from the base model or the memory setting or the training data? This is not analysized sufficiently.

4. The authors could discuss some failing cases.

---

> ### Author Rebuttal · Authors · 2025-07-30
>
> We appreciate your valuable comments, which are crucial for improving our paper! Thank you for finding our "3D benchmark is **novel**" and acknowledging that "the idea of working memory and episodic memory is **interesting**."
>
> > W1.1 & Q1: Efficiency comparison with other methods.
>
> Thank you for this insightful question. We've benchmarked the efficiency on the medium set of our embodied tasks and calculated the average inference time per question, and the results show that our method achieves a **massive improvement** in success rate with only a **marginal increase** in inference time.
>
> **In-domain Tasks**
> | Method | Success Rate ↑ | Time(s)↓ | TFLOPs |
> | :--- | :---: | :---: | :---: |
> | Most Recent Memory | 20.1% | 12.3 | 0.31 |
> | Retrieval-Augmented | 21.8% | 12.1 | 0.36 |
> | 3DLLM-Mem (Ours) | 36.8% | 13.6 | 0.33 |
>
> **In-the-wild Tasks**
> | Method | Success Rate ↑ | Time(s)↓ | TFLOPs |
> | :--- | :---: | :---: | :---: |
> | Most Recent Memory | 12.4% | 13.6 | 0.38 |
> | Retrieval-Augmented | 13.7% | 14.0 | 0.42 |
> | 3DLLM-Mem (Ours) | 31.6% | 14.1 | 0.39 |
>
> The main computational cost for all methods comes from the **7B LLM backbone**. The slight difference in overhead arises from the memory processing step:
>
> * Most Recent Memory: No processing is needed; the memory is fed directly to the LLM.
> * Retrieval-Augmented: Overhead comes from calculating similarity scores against the memory bank.
> * Our Method: Overhead comes from a single cross-attention layer.
>
> The complexity of our fusion step of the query ($N$ x $d$) and past memories key and values which are ($T$ x $N$ x $d$) is $O(TN^2d)$. Since the memory bank size $T$ is bounded by our task definition (Line 142), the $N$ and $d$ shares the same across all methods, this operation is highly efficient. Furthermore, its performance could be easily accelerated using standard optimizations like FlashAttention.
>
> > W1.2: Moreover, if the thinking process is too long, the environment will be changed a lot, and the past observations and interactions might be useless.
>
> Thank you for raising this point. We respectfully suggest that for long-horizon reasoning tasks, **past observations often remain highly relevant**, even after significant time has passed.
>
> For example, in the task shown in Figure 1, the agent must remember the size of the first teddy bear it found to correctly select a matching gift box much later in the episode. If it were to forget this initial information, the task would become unsolvable. The "thinking process" our model performs is precisely what allows it to connect these temporally distant but causally linked events. We are, of course, open to further discussion on scenarios where this might not hold.
>
> > W2: Limited to static environment, while cannot handle the dynamic environment.
>
> Thank you for this concern. We'd like to clarify that our framework is **indeed designed for and evaluated in dynamic environments**. This is a core aspect of our work, mentioned in Lines 3, 7, and 64.
>
> The agent is constantly interacting with and changing the environment by executing actions (as seen in Figures 1 and 3). Our **Memory Update** mechanism (Section 3, Lines 202-209) is specifically designed to handle this. When the agent alters a part of the scene, the memory entry for that location is updated with a new observation, ensuring the model's knowledge of the world stays current.
>
> > W3: Ablation of 3DLLM-Mem effectiveness with base model, the memory setting, the training data.
>
> Our short answer is that the effectiveness comes from our **novel dynamic memory management and fusion module**.
>
> To isolate this contribution, our experimental setup uses the same base model (LLaVA-3D) and the **same training dataset** for all methods. We then compare several memory management techniques: "Everything in Context," "Most Recent Memory," "Retrieval-Augmented Memory," and our proposed 3DLLM-Mem.
>
> As demonstrated in Table 2, our method significantly outperforms all other baselines under these identical conditions. This shows that the performance gain is attributable to our memory fusion design, not the base model or the data. We provide further targeted ablations on the components of our fusion module in Table 3 to reinforce this conclusion.
>
>
> > W4: Failure Cases
>
> For the failure cases, we have identified three common categories:
>
> 1. **Perception Misalignment**: The alignment of the 3D perception encoder with the Large Language Model (LLM) remains a bottleneck in this domain, a challenge we also observed. When a room contains noisy depth data, incomplete 3D meshes, or small/occluded objects, our model can fail to perceive the target objects correctly, leading to incorrect subsequent actions.
> 2. **Premature Task Completion**: We append a ``Task Complete`` token to the end of every training sequence to signal task termination. For complex tasks requiring multiple steps (e.g., collecting four instances of the same object from different rooms), the agent can sometimes lose track of its progress and incorrectly output the ``Task Complete`` token before the goal is fully achieved. We believe employing an explicit "counting" module could help the model track this information, which is an interesting direction for future work.
> 3. **Cascading Errors**: We were pleasantly surprised to find that our model can sometimes recover from an incorrect action mid-task and still successfully complete its goal. However, a more common scenario is that an initial mistake (e.g., picking up the wrong object) leads to a cascade of subsequent incorrect actions. We believe that incorporating reinforcement learning with sparse rewards to penalize incorrect actions could improve model robustness and is another promising area for future research.
>
> > Q2: How to adopt this model into scenes with more dynamics and interaction?
>
> Please see Weakness 2. Our setting is **already dynamic**, and the model is designed to handle interactions that change the environment.
>
> > Q3: What is the contribution of each component in your framework?
>
> Please see Weakness 3. The primary contribution comes from our memory fusion module. We also provide a detailed ablation study in Table 3 that breaks down the effectiveness of its specific design choices.
>
> > Limitation.
>
> We mentioned limitations in our conclusion (Lines 327-330), but we agree this section can be expanded. We will add a more detailed discussion to the appendix in our final version, including the failure cases listed in our response to W4. Thank you for the suggestion.
>
> *We sincerely appreciate your thoughtful comments and hope our responses have addressed your concerns. Please let us know if you have any further questions.*
>
> Best,
> Authors

---

> > ### Author Response · Authors · 2025-08-04
> >
> > Dear Reviewer kteJ,
> >
> > Thank you again for your valuable comments. We hope that our clarifications and additional experiments in the rebuttal have addressed each of your concerns. Should any questions remain unclear, we would appreciate the opportunity for further discussion.
> >
> > Best regards, Authors

---

> ### Comment · Reviewer_kteJ · 2025-08-05
>
> After reading the rebuttal, I think most parts of my concerns have been addressed. However, I still think this work has not provided sufficient evidences for the dynamic scenes, and I hope the authors can address it in the future.
>
> In summary, I keep the positive score.

---

> > ### Author Response · Authors · 2025-08-08
> >
> > Dear Reviewer kteJ,
> >
> > Thank you for your constructive feedback, which is truly encouraging. We deeply appreciate your thoughtful reviews and valuable comments—they have been instrumental in strengthening our paper. We appreciate your engagement throughout this process and are grateful for your valuable contributions.
> >
> > Best regards,
> >
> > Authors

---

### Official Review · Reviewer_VsZD · 2025-06-27

**Clarity:** 2
**Significance:** 4
**Originality:** 3
**Rating:** 4
**Confidence:** 3

**Summary:**

This paper proposes a 3D Vision-Language-Action (3D-VLA) model equipped with long-term memory capabilities. In the context of this work, a 3D-VLA model begins with random exploration, during which it samples scenes randomly and collects RGBD + pose pairs for selected actions (exploration phase). This is followed by task-trajectory execution, where the agent (i) performs actions, (ii) captures corresponding visuals, and iterates until the task is completed or terminated.

The novel contribution of this paper, compared to prior work, is its argument that working memory, commonly used in such agents, is insufficient for tasks requiring reasoning over extended horizons. It further posits that existing benchmarks are too simplistic to evaluate such capabilities effectively.

To address this, the paper introduces a new benchmark based on a synthetic Habitat dataset (covering 182 spaces and 2.6k rooms). Tasks are generated using Gemini, leveraging pre-known scene/object geometry and semantics, and are filtered/verified via a trajectory simulator. A key strength of the dataset is its categorization of trajectories into three difficulty levels (based on the number of rooms required to complete a task) and its division into in-domain and in-the-wild settings (the latter includes areas not seen during model training).

Overall, this benchmark offers a robust testbed for evaluating long-term action planning in indoor environments.

The second contribution is a baseline model that extends LLAVA-3D with long-term memory capabilities. The base network projects patches (tokens) from image/RGBD pairs into the LLM’s embedding space, enabling them to be processed alongside instruction tokens by the base LLM. For spatial reasoning, these embeddings are augmented with 3D positional encodings. To incorporate long-term memory, the model maintains a queue of past observations (termed episodic memory) in addition to the standard LLM context window, enabling retrieval. A memory fusion module allows tokens from active memory to attend to episodic memory, producing a fused representation that is then fed to the base LLM. This enables the LLM to operate not only on active memory but also on context retrieved from long-term storage.

**Questions:**

On the memory update, the paper states:

"The working memory is dynamic and updated online. As the agent interacts with the environment, changes in the environment are immediately reflected in the working memory through updated 3D representations. When the agent moves to a new environment, the previous working memory is transferred to the episodic memory bank. If the corresponding environment already exists in the memory bank and has been modified by the agent, the memory entry is updated accordingly".

Could you please clarify how precisely this is implemented/executed? This appears rather important, I imagine that 'carrying' around full past is in practice not feasible.

Also, please see **Why is the proposed method superior to the Retrieval-Augmented Memory baseline?** above.

**Ethical Concerns:**

["NO or VERY MINOR ethics concerns only"]

**Final Justification:**

The rebuttal addressed my concerns very well, specifically, it clarified the difference between retrieval baseline (a la memorizing transformers) and the proposed methodology (memory fusion), and highlighted conceptually why this approach performs better. Please revise the paper as suggested, especially with regard. crisp problem definition/description.

**Limitations:**

A single sentence in the conclusion: "One limitation of our model is that currently 3DLLM-MEM does not involve low-level navigation and control policy, but utilizes high-level pre-defined policies in simulator for carrying out the actions".

I think this discussion should have been expanded. The success rate of the proposed method is still low (not surprising, the task is hard), but readers would greatly benefit from understanding where the proposed method still fails. Also, at the moment, this model is entirely constrained to synthetic data. There is no evidence for generalization outside of Habitat sandbox. I understand that evaluation outside of that may be difficult, but the bottom line is: no, limitations are not adequately discussed.

**Paper Formatting Concerns:**

None.

**Quality:**

3

**Strengths And Weaknesses:**

###  Strengths

* The paper tackles a significant and underexplored challenge in the 3D-VLM domain: long-term exploration.
* The proposed benchmark is a great contribution and has the potential to advance the field. The benchmark construction process is innovative, particularly with the inclusion of objects from Objaverse (to address the lack of objects in Habitat), alongside LLM-based trajectory generation and simulation-based verification.
* The proposed methodology is a sound adaptation of prior work (LLAVA-3D) and long-term memory mechanisms established in the literature.
* The experimental evaluation is convincing, and I appreciate the inclusion of well-constructed baselines.

### Weaknesses
**Prior work & contextalizing this paper w.r.t prior art.**

The paper effectively discusses related work in the 3D-VLM domain. However, its treatment of long-term memory techniques is incomplete and poorly contextualized. The proposed mechanism appears heavily inspired by existing work in the LLM domain, which extends the "working memory" window by attending to a subset of previously processed data (stored as a KQV queue) retrieved via kNN. For instance, Wu et al.’s “Memorizing Transformers” (ICLR ’22) employs a similar approach (note that the paper states in Section 3.2: “To manage episodic memory, we propose the use of a memory feature bank,” whereas it should acknowledge following established practice).

While extending this concept to the 3D-VLM domain is a notable contribution, the omission of this discussion is concerning.


**Presentation:**

1. Certain sections of the paper are well-written, particularly those drawing parallels between memory modules and neuroscience, which I appreciate. However, the paper falters in addressing fundamental questions: What is the task at hand? What are the model’s inputs and outputs? What is the problem statement or task definition? The paper is so focused on the “how” that it fails to clearly communicate the “what.” While this information can be pieced together from the paper or related works, its absence in the introductory section places an unnecessary burden on the reader. I only fully understood the task definition in the “Embodied Data Collection” paragraph of Section 2.2 (Data Collection).

2. The paper builds upon LLAVA-3D and augments it with long-term memory capabilities. To be self-contained, it should include a preliminary section briefly describing the LLAVA-3D architecture. Additionally, the paper would benefit from a clear problem setting and task definition section before delving into data collection, generation, and modeling, with these elements foreshadowed in the introduction.

**Why is the proposed method superior to the Retrieval-Augmented Memory baseline?**
The aforementioned baseline seems very similar to the proposed method, yet experimentally, the proposed model performs significantly better. It is unclear from the experimental evaluation where this difference comes from, as well as what precisely is the key difference at the methodology level responsible for the performance gains. From the paper description, it appears to me that the key difference is in how the memory fusion is performed?

**Nit:**
* Abstract: “reason over long-term memory in 3D environments” → consider rephrasing to “reason over extended temporal spans using long-term memory” for clarity.
* Table 1 analyzes different datasets and includes a column labeled “Long-Term Memory.” Benchmarks do not possess memory, so this label is misleading.


I am genuinely excited about this work, but its current presentation issues are significant. I am open to revising my rating upward after the rebuttal if my concerns are adequately addressed.

---

> ### Author Rebuttal · Authors · 2025-07-30
>
> We'd like to express our sincere gratitude for your thorough review of our paper. We're thrilled that you found our work **"tackles a significant and underexplored challenge,"** that our **"benchmark is a great contribution,"** and that our **"experimental evaluation is convincing."** We greatly appreciate your suggestions, which are crucial for improving the quality of our paper.
>
> > W1: Contextalizing our work w.r.t prior work.
>
> Thank you for pointing out “Memorizing Transformers” (ICLR ’22). While our initial literature survey focused on embodied AI, we agree this is a relevant work and will cite and discuss it in our revised manuscript. Its approach highlights **key differences that underscore the novelty of our method**:
>
> 1. **Memery Fusion vs. Retrieval**: “Memorizing Transformers” utilizes a k-NN lookup to retrieve top (key, value) pairs from memory. This is a form of **retrieval**, similar in principle to our "Retrieval-Augmented Memory" baseline. Our proposed 3DLLM-Mem method is **fundamentally different**; it uses **fusion**. While retrieval selects a subset of the "best" memories based on a fixed similarity score, our fusion module **attends to all memories simultaneously**. It uses a parameterized cross-attention mechanism (Line 196) to learn a new, task-relevant representation by selectively aggregating information from the entire memory bank. This allows the model to learn what past information is most important for the current step, a capability our experiments show is highly effective.
> 2. **Dynamic vs. Static Memory**: “Memorizing Transformers” stores a long-term context (like a text document) as a static KV cache. In our embodied setting, the memory is dynamic. As the agent interacts with the environment, the world state changes (Line 202, Memory Update). Our model is designed to handle this dynamism by continuously updating its memory bank and fusing these updated memories at each step to inform its actions, as shown in Figure 3.
> 3. **Different Problem Settings**: As you acknowledged, the settings are distinct. “Memorizing Transformers” focuses on extending the effective context window for language models on text-based tasks. Our work, as stated in our abstract, introduces "a novel dynamic memory management and fusion model for embodied spatial-temporal reasoning and actions."
>
> Moreover, these **significant differences in application settings fundamentally shape model design choices**—for example, the approach to extending a language transformer’s context window via KV cache versus enabling spatial-temporal memory for dynamic 3D embodied tasks.
>
> > W2: Presentation: What is the task definition and what are the model’s inputs and outputs? The 'what' is missed from paper's Introductory section.
>
> Thank you for highlighting this. We apologize if the introduction was not perfectly clear. We will definitely revise the introduction to be more explicit.
>
> To clarify here:
>
> * The task is for an embodied agent, given a high-level language instruction (e.g., "prepare the most suitable gift box for the teddy bear."), to navigate a 3D environment and perform a sequence of actions to complete the goal. We provide a visual "teaser" of this in Figure 1.
> * The model's input at each step is the current 3D visual observation (working memory) and a bank of past 3D observations (episodic memory). Please also refer to paper's Sec 1, Line 71.
> * The model's output is a symbolic action, such as PickUp(teddy bear).
>
> We will incorporate this clear problem definition into the introductory section of our revised paper.
>
>
> > W3: Presentation: A preliminary section briefly describing the LLAVA-3D and clear problem setting and task definition section before Section 2.
>
> Thank you for this excellent structural suggestion. We'd like to clarify that we do introduce LLaVA-3D as a preliminary in **Section 3.1 (Lines 165-176)**.
>
> However, we agree with your point about the overall structure. Our original intention was to present our benchmark (which includes the task settings) before the model. Based on your feedback, we see that introducing the **high-level problem setting and task definition _before_ Section 2** would improve the paper's flow. We will make this structural change in our final version.
>
> > W4 & Q2: Why is the proposed method superior to the Retrieval-Augmented Memory baseline?
>
> You are exactly right that "the key difference is in how the memory fusion is performed." As we elaborated in our response to W1, the core distinction is **Fusion vs. Retrieval**.
>
> To summarize: The Retrieval-Augmented baseline selects the top-K "best" memories based on a static similarity metric. This has two potential drawbacks: the similarity metric may not be optimal, and the top-K memories might not contain all the necessary context.
>
> In contrast, our fusion-based method **attends over all memories** and learns a **dynamic, task-dependent** way to combine their information. It isn't limited to a few retrieved memories; it can synthesize information from the entire history. This allows it to capture more complex, long-range dependencies and makes it more robust, as it's not dependent on a single, potentially flawed retrieval step.
>
>
> > Reponse to Nit:
>
> Thank you so much for your careful reading! We will happily make these changes for clarity in the revised version:
>
> * We'll change “reason over long-term memory in 3D environments” to “reason over extended temporal spans using long-term memory.”
> * We'll change the column name "Long-term Memory" in Table 1 to something less ambiguous, such as "Evaluates Memory," to avoid confusion.
>
> > Q1: Clarify how precisely memery update is implemented/executed?
>
> We are happy to further explain the memory update process (Lines 202-209).
>
> First, as noted in Line 142, the number of distinct locations (e.g., rooms) in any task is finite, so the memory bank doesn't grow indefinitely. The memory bank itself is a structured collection of feature vectors, where each vector corresponds to a specific, previously visited location.
>
> When the agent revisits a location and performs an action that alters the scene (e.g., picking up an object), we re-compute the 3D feature representation for that location. We then **update the corresponding entry in the memory bank** with this new feature vector. Implementation wise, it's as simple as Bank(entry)=new_vector. This is what's described in Lines 205-207: "If the corresponding environment already exists in the memory bank and has been modified by the agent, the memory entry is updated accordingly." This process ensures the agent's memory of the world state remains current.
>
>
> > Limitation:
>
>
> Thank you for this valuable suggestion. We did have to shorten our limitation section due to space constraints, but we will add a more comprehensive discussion to the Appendix, including real-world evaluation and a detailed breakdown of failure cases.
>
> As a preview, here are the three common failure categories we've identified:
>
> 1. **Perception Misalignment**: The alignment of the 3D perception encoder with the Large Language Model (LLM) remains a bottleneck in this domain, a challenge we also observed. When a room contains noisy depth data, incomplete 3D meshes, or small/occluded objects, our model can fail to perceive the target objects correctly, leading to incorrect subsequent actions.
> 2. **Premature Task Completion**: We append a ``Task Complete`` token to the end of every training sequence to signal task termination. For complex tasks requiring multiple steps (e.g., collecting four instances of the same object from different rooms), the agent can sometimes lose track of its progress and incorrectly output the ``Task Complete`` token before the goal is fully achieved. We believe employing an explicit "counting" module could help the model track this information, which is an interesting direction for future work.
> 3. **Cascading Errors**: We were pleasantly surprised to find that our model can sometimes recover from an incorrect action mid-task and still successfully complete its goal. However, a more common scenario is that an initial mistake (e.g., picking up the wrong object) leads to a cascade of subsequent incorrect actions. We believe that incorporating reinforcement learning with sparse rewards to penalize incorrect actions could improve model robustness and is another promising area for future research.
>
> *We sincerely appreciate your thoughtful comments and hope our responses have addressed your concerns. Please let us know if you have any further questions.*
>
> Best,
> Authors

---

> > ### Author Response · Authors · 2025-08-04
> >
> > Dear Reviewer VsZD,
> >
> > Thank you again for your valuable comments. We hope that our clarifications in the rebuttal have addressed each of your concerns. Should any questions remain unclear, we would appreciate the opportunity for further discussion.
> >
> > Best regards, Authors

---

> > ### Comment · Reviewer_VsZD · 2025-08-05
> >
> > Thank you for a great response; the rebuttal addressed my concerns very well. I am upgrading my rating to borderline accept.

---

> > > ### Author Response · Authors · 2025-08-08
> > >
> > > Dear Reviewer VsZD,
> > >
> > > Thank you for your thoughtful review. We were truly encouraged by your supportive assessment of our work. Your feedback on the paper's flow and clarity was invaluable, and we will certainly incorporate your suggestions into the final version. We are grateful for your engagement and significant contributions to improving our paper.
> > >
> > > Best regards,
> > >
> > > The Authors

---

### Official Review · Reviewer_9Co4 · 2025-06-30

**Clarity:** 2
**Significance:** 3
**Originality:** 3
**Rating:** 5
**Confidence:** 4

**Summary:**

This paper focuses on the challenge of 3D spatial understanding and memory retrieval in 3D-LLMs. To that end, the authors propose 3DLLM-Mem, a 3D LLM that features a novel memory fusion mechanism inspired by human memory and cognitive maps. In particular, the model uses working memory and episodic memory tokens to represent current and past observations and retrieve relevant memories.

To evaluate this method and support other research in this direction, the authors also introduce 3DMem-Bench, a benchmark with trajectories and tasks spanning embodied navigation, question answering, and captioning. They benchmark their method, along with recent 3D-LLMs and memory mechanisms, and show that their method significantly outperforms prior work on all tasks.

**Questions:**

1. What are the core differences between in-domain and in-the-wild tasks?
2. How are the following terms defined: instance, environment, unseen memory context?
3. What is the relationship between the working and episodic memory, and how are observations transferred from one to the other? Does the working memory contain observations from more than a single timestep, and if so, how is a new environment (as mentioned in lines 204-205) determined?
4. Do you have insight into why 3DLLM-Mem demonstrates strong generalization capabilities? Is this simply due to better overall performance? Or are there specific components of the method that enable it to maintain performance compared to other approaches?

**Ethical Concerns:**

["NO or VERY MINOR ethics concerns only"]

**Final Justification:**

The authors have addressed my concerns about the clarity of the problem setup and generalization contribution, showing that they've made significant improvements compared to prior works by highlighting the relative performance drop as a metric and providing high-level insight into how their method is capable of generalizing. I've chosen to raise my score to a 5.

**Limitations:**

Yes

**Quality:**

3

**Strengths And Weaknesses:**

**Strengths**

- Proposed memory fusion mechanism is novel and well-justified
- The proposed 3DMem-Bench benchmark is the first benchmark focusing on long-term memory and fine-grained complexity, and good details about the benchmark design principles and dataset collection process are provided
- Strong results on both navigation and embodied QA tasks, demonstrating significant improvement over prior works
- Demonstrates strong generalization to unseen environments

**Weaknesses**

That said, I have some concerns about the clarity of the method and generalization contribution.
1. Throughout the paper, there are some terms that are not properly defined and missing details, which make parts of the method unclear. In particular:
    - It’s unclear what the difference is between the in-domain vs in-the-wild tasks. A better understanding of this would be helpful for understanding the generalization capabilities of 3DLLM-Mem.
         - The details in Section 2.3 about how the in-domain and in-the-wild datasets are collected are highly specific, which makes it harder to understand what the core difference is between in-the-wild and in-domain tasks.
         - Also, this section references “instances” and “unseen memory context”, which are not defined earlier and also make the description unclear
    - The definitions of the environment, working memory, and episodic memory are unclear
         - I’m confused about what observations are part of the working versus episodic memory. In line 182, it says that the current observation at each time step is stored in working memory, implying that the working memory stores a single timestep. In lines 204-205 though, it seems like the working memory is longer since it explains that “When the agent moves to a new environment, the previous working memory is transferred to the episodic memory bank.”
         - It would help to have a concrete formulation of these terms, as they’re core to the method
    - Details about training are missing.

2. The paper emphasizes that 3DLLM-Mem demonstrates strong generalization to unseen environments. While 3DLLM-Mem has much higher absolute performance on in-the-wild tasks, the relative drop in performance between in-domain and in-the-wild is only slightly lower than other methods. More insight into this would strengthen this contribution, such as explaining which components of the method lead to this improvement.

---

> ### Author Rebuttal · Authors · 2025-07-30
>
> Thank you for your careful reading and detailed feedback. We're glad you found our proposed memory fusion mechanism to be "**novel and well-justified**", and we appreciate you highlighting our model's "**strong results and generalization to unseen environments**".
>
> We provide detailed explanations for the points of clarity you raised below and will update our paper accordingly to ensure these concepts are easier to understand. We're also happy to engage further during the discussion period to clarify any remaining questions.
>
> > W1 & Q1: What the difference is between the in-domain vs in-the-wild tasks? What are “instances” and “unseen memory context”?
>
> Thank you for this question. Let's first clarify the terms to facilitate the explanation.
>
> * Instance: An "instance" is simply a single data point or evaluation episode.
> * Environment: An "environment" refers to a specific 3D scene, such as a particular house layout with its rooms and objects. As illustrated in Figure 1, the agent navigated from the bedrooms to the living room, and then to the kitchen—all of which constitute the environment.
> * Unseen Memory Context: This means the agent is operating in an environment that it has never encountered during training.
>
> With these definitions, the distinction between our evaluation splits (please also refer to paper's Lines 144-148) is as follows:
>
> 1. **From an Environment/Memory Perspective**:
> * In-domain tasks take place in environments that may have been seen during training. Therefore, the agent's episodic memory bank might contain memories from a familiar scene. However, the agent's current observation (its working memory) is always from a novel, unseen state within that environment.
> * In-the-wild tasks, as noted in Line 147, feature **"entirely unseen memory context."** This means the agent operates in completely new environments that were not part of the training data.
>
> 2. **From a Task Difference Perspective**:
> * In-domain tasks follow the same distribution and types of tasks as those in the training set. For example, as illustrated in Figure 2, one such task is "Find an ideal container to store all the cookie dough."
> * In-the-wild tasks, as noted in Line 148, introduce **novel challenges** that are more complex and qualitatively different from those seen during training. Examples of these challenging tasks are provided in Figure 2; corresponding to the in-domain example above, for instance, is "Collect all cookie dough and then **rearrange** it on the living room table in descending order of size."
>
> *A brief note on design*: We designed the in-domain set this way to ensure the in-the-wild set remained a reasonably sized and robust evaluation pool containing our most valuable, entirely novel instances. The in-domain tasks are still challenging, as shown by the experimental results, because the agent must still act based on its immediate, unseen working memory.
>
>
> > W2: The definitions of the environment, working memory, and episodic memory are unclear.
>
> We apologize for the confusion regarding the terms "working memory" and "episodic memory," which are inspired by concepts in neuroscience. Your understanding that **"working memory stores a single timestep"** is exactly right! It represents the agent's perception of its current 3D environment at a specific moment.
>
> We believe the confusion may have stemmed from the sentence: "When the agent moves to a new environment, the previous working memory is transferred to the episodic memory bank." To be more precise: at each timestep $t$, the agent's current observation is its working memory. When the agent proceeds to timestep $t$+1, the observation from the previous step $t$ becomes a part of its past experience. This past observation is then incorporated into the episodic memory bank. This process is visualized in Figure 3(b), where the working memory tokens $f_t^Q$ are fused with the past memory bank features.
>
> To further emphasize this distinction, **our paper defines** their structures with mathematical notation:
>
> * Working Memory (Line 182) has a shape of $R^{(N \times d)}$, representing features from one time step.
> * Episodic Memory (Line 184) has a shape of $R^{(T \times N \times d)}$, representing features from $T$ past time steps.
>
> We hope this resolves the confusion, and we will revise the text in the paper to make this relationship clearer
>
> > W3: Details of training are missing.
>
> We appreciate you raising this, and we would like to clarify where these details are located in the paper. We describe our training setup in **Section 4.1**, under the paragraph **Implementation Details** (Lines 218-224). This section covers the training computation, architecture, and loss function.
>
> For more granular details, we mention in Line 224 that a full list of hyperparameters is available in **Appendix D** (Lines 691-699). There, we specify the learning rate, optimizer, weight decay, and learning rate scheduler used for our experiments. We will also open-source our code after the review period. Thank you for your understanding.
>
> > W4.1: While 3DLLM-Mem has much higher absolute performance on in-the-wild tasks, the relative drop in performance between in-domain and in-the-wild is only slightly lower than other method.
>
> Thank you for this insightful observation. To better address your point, we've created a new table using the scores from Table 2, adding columns for absolute and relative performance drops.
>
> ||In-domain SR|In-the-wild SR|Drop (Absolute)|Drop (Relative)|
> -|-|-|-|-|
> Most Recent Memory|21.1|13.7|-7.4|-35.1%|
> Retrieval-Augmented Memory|22.3|15.6|-6.7|-30.0%|
> 3DLLM-Mem (Ours)|37.6|32.1|-5.5|-14.6%|
>
>
> As illustrated above, our method demonstrate a **-5.5** absolute and a **-14.6%** relative performance drop which is significantly smaller than the >30% relative drop seen with other methods. We believe this demonstrates a substantial improvement in generalization, not just a "slight" one.
>
> Furthermore, we'd argue that relative drop is a more equitable metric here. It's inherently more challenging to **maintain a high level of performance** (like our 37.6% -> 32.1%) on difficult, unseen tasks than it is to **maintain a low level of performance** (like 21.1% -> 13.7%). Our model's ability to retain most of its high performance highlights its robustness.
>
> > W4.2: More insight into this would strengthen this contribution, such as explaining which components of the method lead to this improvement.
>
> Thank you for recognizing our method's effectiveness. We attribute this strong generalization capability directly to our **novel memory fusion module** (Section 3.2), which excels for two key reasons:
>
> 1. **Relevant Initialization**: The fusion process begins with query tokens initialized from the agent's current working memory. This is crucial because it primes the attention mechanism to selectively retrieve and focus on past memories that are most relevant to the agent's immediate surroundings and task. As shown in our ablation study (Table 3), initializing from other sources leads to inferior performance.
> 2. **Holistic Fusion Process**: Unlike simple retrieval methods that might only fetch the single "best" memory, our fusion module **attends to all past memories simultaneously**. It learns to weigh and synthesize them into a single, compact, and context-rich representation. This allows the model to capture complex, long-term dependencies and make more informed decisions, which is especially critical for the novel and challenging tasks in the in-the-wild set.
>
> > Q2: How are the following terms defined: instance, environment, unseen memory context?
>
> Please see Weakness 1 and 2.
>
> > Q3: What is the relationship between the working and episodic memory?
>
> Please see Weakness 2. In short, working memory is the agent's current observation, which becomes part of the episodic (past) memory in the next timestep.
>
> > Q4: Do you have insight into why 3DLLM-Mem demonstrates strong generalization capabilities? Is this simply due to better overall performance? Or are there specific components of the method that enable it to maintain performance compared to other approaches?
>
> Please see Weakness 4.2. The performance improvement is from our deisgned fusion memory module as introduced in paper's section 3.2.
>
> *We sincerely appreciate your thoughtful comments and hope our responses have addressed your concerns. Please let us know if you have any further questions.*
>
> Best,
> Authors

---

> > ### Comment · Reviewer_9Co4 · 2025-08-04
> >
> > Thank you for addressing my concerns about the clarity of the problem setup. In particular, the results highlighting the relative performance drop as a metric help demonstrate the improved generalization capabilities of 3DLLM-Mem compared to prior works. I've updated my rating accordingly.

---

> > > ### Author Response · Authors · 2025-08-04
> > >
> > > Thank you for your timely and constructive feedback, which is truly encouraging. We deeply appreciate your thoughtful reviews and valuable comments—they have been instrumental in strengthening our paper. It was a pleasure engaging in this discussion, and we are glad that we could address your concerns.
> > >
> > > Best regards,
> > >
> > > Authors

---

### Official Review · Reviewer_syxz · 2025-07-05

**Clarity:** 3
**Significance:** 3
**Originality:** 3
**Rating:** 4
**Confidence:** 2

**Summary:**

This paper introduces a novel benchmark 3DMEM-BENCH for reasoning, planning and acting with long-term spatial-temporal memory in embodied environments. In the meantime ,this paper introduces  a 3D embodied LLM with dynamic memory management capabilities designed specifically for embodied spatial-temporal reasoning, planning and acting, and evaluates popular 3D-LLMs and memory mechanisms on 3DMEM-BENCH.

**Questions:**

Section 2.2 notes a low 24% validation rate for Gemini-generated trajectories. What were the most common failure modes that caused these trajectories to be invalid (e.g., illegal navigation, violations of object interaction rules (such as the "hold only one object at a time" constraint mentioned in Appendix F.1)? Could you provide several failure case analysis?

**Ethical Concerns:**

["NO or VERY MINOR ethics concerns only"]

**Final Justification:**

The author's response addressed most of my concerns. I will maintain a positive score.

**Limitations:**

yes

**Quality:**

3

**Strengths And Weaknesses:**

Strengths
1. This paper collected a fine-grained benchmark with multiple difficulty levels and varying numbers of multi room scene settings, which can assist the model in evaluation.
2. This paper proposes a new mechanism for modeling long-term spatio-temporal memory, which interacts with attention and historical memory to effectively improve the performance of the model in complex scenarios.
3. The approach proposed in the paper when switching environments involves storing working memory in the episode memory bank, which greatly reduces costs through this mode of thinking.

Weaknesses
1. The paper describes how memories are added to and updated in the episodic memory bank, but not how they might be pruned or consolidated. In extremely long-horizon scenarios, this could lead to the information redundancy and the memory bank growing indefinitely, which might reintroduce the very problems of computational inefficiency.
2. While a significant effort was made to validate the data , tasks generated by an LLM (Gemini)  and executed within a simulator may not fully capture the physical world. This creates a risk that the model may be learning to exploit artifacts of the simulation rather than developing skills that are robustly transferable to real-world problems.

---

> ### Author Rebuttal · Authors · 2025-07-30
>
> We sincerely thank you for your constructive feedback and for describing 3DMem-Bench as a **"novel and fine-grained"** benchmark and 3DLLM-Mem "**effectively improves the performance of the model in complex scenarios**". We greatly appreciate your suggestions which are crucial in improving the quality of our paper.
>
> >W1: How to handle scenarios when memory bank growing indefinitely?
>
> Thank you for raising this valuable concern. In the current benchmark configuration (Sec.2, L.142), an embodied task involves at most 10 rooms. Yet we have constructed a challenging benchmark as all models' performance is still low. Therefore, in the context of 3DMem-Bench,  the memory bank **remains small enough to avoid computational inefficiency**.
>
> Nevertheless, we agree that scalability is an essential consideration for broader real-world applications. We are happy to discuss the following promising approaches to manage memory growth:
>
> 1.  Sliding-window memory (FIFO). Maintain only the most recent N memories and discard older ones. This is similar to the “Most-Recent Memory” baseline in our paper but now applies to the memory bank. This method keeps both latency and memory O(N).
> 2.  Memory consolidation / compression. The cross attention complexity of the query ($N$ x $d$) and past memories key and values which are ($T$ x $N$ x $d$) is $O(TN^2d)$.
> We can: 1) down-sample token length $N$. 2) project to a lower dimension $d$ (e.g., LoRA-style adapters), or 3) cluster similar memories and retain only their representative centroids. While promising, a careful study of the resulting accuracy–efficiency trade-off would be an excellent topic for future work.
> 3. Retrieval-augmented subsets.
> Index memories with a light-weight key–value store (FAISS, ScaNN) and then apply memory fusion only to the top-K retrieved memories. This reduces the effective $T$ and is orthogonal to the above techniques.
>
> We believe all these approaches can be promising future works but **beyond the scope of the current paper**.
>
> >W2: The tasks in a simulator may not fully capture the physical world.
>
> Thank you for this feedback. We share this concern and directly investigated the generalization and transferability of our model within the 3DMem-Bench by splitting tasks into "in-domain" and "in-the-wild" sets. Our results show that 3DLLM-Mem demonstrates **strong generalization** capabilities on the unseen "in-the-wild" tasks.
>
> Moreover, we believe our approach has **high potential for real-world transferability**. As illustrated in paper's Figure 1 and 3, the model outputs high-level, symbolic actions ``(e.g., Navigate(room=kitchen), PickUp(bottle))``. The low-level motion primitives are executed by modules that are already deployed and tested on real robots. Consequently, the learned policy primarily needs to decide *what to do*, not *how to do it*, which substantially narrows the reality gap often encountered in methods that learn low-level continuous control policies.
>
> >Q1: Failure Cases.
>
> Thank you for this question. We'll clarify the failure cases for both the data generation process and our trained model.
>
> Regarding the **task generation process**, as you noted, we discuss these limitations in Appendix F.1. The primary failure modes involve Gemini occasionally generating trajectories that violate constraints related to: 1) the agent’s final location, 2) the specific object referenced, and 3) the correctness of pick-up and put-down actions. The single most common failure is violating the "hold only one object at a time" constraint, as this requires strong reasoning and a consistent memory of the agent's current inventory.
>
> For the failure cases of our trained **3DLLM-Mem model**, we have identified three common categories:
>
> 1. **Perception Misalignment**: The alignment of the 3D perception encoder with the Large Language Model (LLM) remains a bottleneck in this domain, a challenge we also observed. When a room contains noisy depth data, incomplete 3D meshes, or small/occluded objects, our model can fail to perceive the target objects correctly, leading to incorrect subsequent actions.
> 2. **Premature Task Completion**: We append a ``Task Complete`` token to the end of every training sequence to signal task termination. For complex tasks requiring multiple steps (e.g., collecting four instances of the same object from different rooms), the agent can sometimes lose track of its progress and incorrectly output the ``Task Complete`` token before the goal is fully achieved. We believe employing an explicit "counting" module could help the model track this information, which is an interesting direction for future work.
> 3. **Cascading Errors**: We were pleasantly surprised to find that our model can sometimes recover from an incorrect action mid-task and still successfully complete its goal. However, a more common scenario is that an initial mistake (e.g., picking up the wrong object) leads to a cascade of subsequent incorrect actions. We believe that incorporating reinforcement learning with sparse rewards to penalize incorrect actions could improve model robustness and is another promising area for future research.
>
> *We sincerely appreciate your thoughtful comments and hope our responses have addressed your concerns. Please let us know if you have any further questions.*
>
> Best,
> Authors

---

> > ### Author Response · Authors · 2025-08-04
> >
> > Dear Reviewer syxz,
> >
> > Thank you again for your valuable comments. We hope that our clarifications in the rebuttal have addressed each of your concerns. Should any questions remain unclear, we would appreciate the opportunity for further discussion.
> >
> > Best regards, Authors

---

> > ### Comment · Reviewer_syxz · 2025-08-06
> >
> > Thank you for your response, which has addressed most of my concerns. I will maintain a positive score.

---

> > > ### Author Response · Authors · 2025-08-08
> > >
> > > Dear Reviewer syxz,
> > >
> > > Thank you for your thoughtful review. We were truly encouraged by your supportive assessment of our work. Your constructive feedback was invaluable, and we will certainly incorporate your suggestions into the final version. We are grateful for your engagement and significant contributions to improving our paper.
> > >
> > > Best regards,
> > >
> > > The Authors

---

### Comment · Area_Chair_oGgp · 2025-08-02
**Post-rebuttal Discussions and Finalizing Scores**

Dear Reviewers,

This paper receives mixed ratings before rebuttal (3 borderline accepts and 1 borderline reject). Please read the author rebuttals, actively discuss with the authors before August 6, and finalize your scores before August 13.

Thank you for your contribution to the reviewing process!

Best regards,
Area Chair

---

### Author Response · Authors · 2025-08-08
**Summary of Rebuttal**

We sincerely thank all the reviewers for their time and their constructive reviews and suggestions. We are encouraged that the reviewers find that:

* Our 3DMem-Bench a **novel and fine-grained** benchmark (Reviewers syxz, 9Co4, VsZD, kteJ), is a **great contribution** and has the potential to advance the field (Reviewer VsZD).
* Our method is **novel and well-justified** (Reviewers syxz, 9Co4), **interesting** (Reviewer kteJ), and **tackles a significant and underexplored challenge** in the 3D-VLM domain: long-term exploration (Reviewer VsZD).
* The experimental evaluation is **convincing** with inclusion of well-constructed baselines (Reviewer VsZD), which demonstrates our method **effectively** and **significantly** improves the model performance in complex scenarios over prior works (Reviewers syxz, 9Co4) and has **strong generalization to unseen environments**. (Reviewer 9Co4)

Throughout the discussion period, we have worked diligently to address all reviewer concerns. Key clarifications and additions include:

* **Methodology & Novelty**: We clarified our core Memory Fusion mechanism versus retrieval-based methods, explained the dynamic memory update process, and contextualized our work against the suggested prior art (Reviewers syxz, VsZD).
* **Benchmark & Evaluation**: We clarified the distinction between in-domain vs. in-the-wild tasks and working vs. episodic memory. We also added a new analysis of the relative performance drop to further demonstrate our model's superior generalization (Reviewer 9Co4).
* **Efficiency & Practicality**: We provided a new efficiency analysis to show our method's competitive performance, addressed concerns about memory growth, and clarified that our method is already designed for dynamic environments (syxz, kteJ).
* **Failure Cases & Limitations**: We provided a transparent analysis of our model's failure modes (Perception Misalignment, Premature Completion, and Cascading Errors) and have committed to expanding the limitations and ethics sections in the final manuscript (syxz, VsZD, kteJ).

We have provided detailed responses to each reviewer individually. We believe the resulting discussion and revisions have substantially improved the clarity and impact of our paper.

Thank you once again for your valuable contributions to this process. We believe the comments and revisions have made the paper stronger.

---

### Decision · Program_Chairs · 2025-09-17

**Decision:**

Accept (poster)

**Comment:**

This paper proposes 3DMem-Bench designed to evaluate an agent's ability to reason over long-term memory in 3D environments, and 3DLLM-Mem, a novel dynamic memory management and fusion model for embodied spatial-temporal reasoning and actions in LLMs. Reviewers acknowledge the strengths of the paper, including the importance of the problem, the novelty and significance of the approach, and the convincing experiments. Reviewers raised concerns about generalizability to real-world scenes, clarity of evaluation setting, computation cost, comparison with other approaches, etc. Most questions are effectively addressed by authors during the rebuttal, except for one reviewer still having concerns on dynamic scenes. Despite this, all reviewers agreed on positive scores after rebuttal and recognize the merits of the work. After carefully considering the paper, reviewer comments, and rebuttal, the area chair recommends accepting this paper.